# Metabolic control of CD47 expression through LAT2-mediated amino acid uptake promotes tumor immune evasion

Zenan Wang [1,2,3,7], Binghao Li[1,2,3,7], Shan Li[4,7], Wenlong Lin[1,5], Zhan Wang[1,2,3], Shengdong Wang[1,2,3], Weida Chen[1], Wei Shi[1], Tao Chen[1,2,3], Hao Zhou[1,2,3], Eloy Yinwang[1,2,3], Wenkan Zhang[1,2,3], Haochen Mou[1,2,3], Xupeng Chai[1,2,3], Jiahao Zhang[1,2,3], Zhimin Lu[4,6] & Zhaoming Ye [1,2,3] ✉

Chemotherapy elicits tumor immune evasion with poorly characterized mechanisms. Here, we demonstrate that chemotherapy markedly enhances the expression levels of CD47 in osteosarcoma tissues, which are positively associated with patient mortality. We reveal that macrophages in response to chemotherapy secrete interleukin-18, which in turn upregulates expression of L-amino acid transporter 2 (LAT2) in tumor cells for substantially enhanced uptakes of leucine and glutamine, two potent stimulators of mTORC1. The increased levels of leucine and enhanced glutaminolysis activate mTORC1 and subsequent c-Myc-mediated transcription of CD47. Depletion of LAT2 or treatment of tumor cells with a LAT inhibitor downregulates CD47 with enhanced macrophage infiltration and phagocytosis of tumor cells, and sensitizes osteosarcoma to doxorubicin treatment in mice. These findings unveil a mutual regulation between macrophage and tumor cells that plays a critical role in tumor immune evasion and underscore the potential to intervene with the LAT2-mediated amino acid uptake for improving cancer therapies.

Osteosarcoma is the most common primary malignant tumor of the bone. The introduction of chemotherapy since 1970s has dramatically improved the overall survival of osteosarcoma patients. At present, radical surgery combined with neoadjuvant and adjuvant chemotherapy has been standardized for osteosarcoma treatment[1]. A combination of cisplatin, doxorubicin, methotrexate, and ifosfamide has been used as chemotherapy regimens for decades[2]. Despite the significantly improved efficacy of chemotherapy, the clinical prognosis of osteosarcoma remains unsatisfactory. About one-third of osteosarcoma patients respond poorly to chemotherapy drugs, who are prone to develop recurrent and metastatic diseases, with their 5-year survival rates at only 5–20%[3]. Chemoresistance and immune evasion have become major challenges in osteosarcoma treatment, and more effective strategies are needed to improve patient survival[4].

The ability of cancer cells to evade from immune surveillance plays a critical role in cancer relapse and metastasis[5,6]. Though chemotherapeutic agents eradicate tumor cells in part by stimulation of the immune system[7], growing evidence have revealed that tumors following chemotherapy acquire an enhanced resistance to antitumor immunity[8,9]. Macrophages are the most abundant immune cell type in osteosarcoma[10,11], and the association between their density and clinical outcome remains contradictory[12–15]. Analysis of pre-chemotherapy

[1]Department of Orthopedic Surgery, The Second Affiliated Hospital, Zhejiang University School of Medicine, Hangzhou, Zhejiang, China. [2]Orthopedics Research Institute of Zhejiang University, Hangzhou, Zhejiang, China. [3]Key Laboratory of Motor System Disease Research and Precision Therapy of Zhejiang Province, Hangzhou, Zhejiang, China. [4]Department of Hepatobiliary and Pancreatic Surgery and Zhejiang Provincial Key Laboratory of Pancreatic Disease of The First Affiliated Hospital, Institute of Translational Medicine, Zhejiang University School of Medicine, Hangzhou, Zhejiang, China. [5]Institute of Immunology, Zhejiang University School of Medicine, Hangzhou, Zhejiang, China. [6]Zhejiang University Cancer Center, Hangzhou, Zhejiang, China. [7]These authors contributed equally: Zenan Wang, Binghao Li, Shan Li. ✉e-mail: yezhaoming@zju.edu.cn

osteosarcoma specimens have revealed that increased macrophage density was related to metastasis suppression and longer metastasis progression-free survival[13,14], while studies of post-chemotherapy specimens showed osteosarcoma patients with detectable metastasis had more macrophages in tumors compared to those without metastasis[15]. These results suggest an enhanced evasion of macrophage immunosurveillance by cancer cells during osteosarcoma chemotherapy through some unknown mechanisms.

A major mechanism that mediates evasion of innate immunity by cancer cells is the expression of CD47, which is a potent macrophage immune checkpoint[16]. CD47 on tumor cells can bind to its ligand, signal regulatory protein α (SIRPα), on macrophages, thereby inhibiting macrophage phagocytosis[16]. In osteosarcoma, CD47 is usually expressed at higher levels in the cancer cells than in the surrounding normal tissue[17,18]. Higher CD47 expression is associated with worse survival in osteosarcoma patients[19]. Activating tumor-associated macrophages by inhibiting CD47 augmented the macrophage-mediated clearance of tumor cells and inhibited osteosarcoma growth and pulmonary metastases in mice[17,20,21]. CD47 blockade has shown promising activity in patients with advanced hematologic cancers[22], and clinical trials are currently being designed to evaluate whether CD47 inhibition enhances the activity of other immunotherapies in osteosarcoma patients[18]. However, whether CD47 expression in osteosarcoma cells is regulated in response to chemotherapy, thereby promoting tumor immune evasion, is unknown.

Cancer cells demand an adequate supply of nutrients to sustain their rapid growth and proliferation. Amino acid transporters are vital in maintaining a high level of metabolism and protein synthesis in cancer cells[23]. Therefore, enhanced amino acids uptake through upregulation of specific transporters has been observed in many primary human tumors, as well as in various cancer cell lines[24]. The L-type amino acid transporters (LATs) make up a major Na$^+$-independent transporting system for neutral amino acids, such as glutamine and leucine[25]. So far, four LATs (LAT1, LAT2, LAT3, and LAT4) have been identified, and blocking LAT1 has become an attractive strategy in cancer therapy[25]. However, the relationship between amino acids uptake and regulation of tumor immune checkpoints is unclear.

In this study, we find that CD47 is upregulated in osteosarcoma in response to chemotherapy, and this upregulation is associated with patient mortality. Mechanistically, chemotherapy promotes macrophage secretion of interleukin (IL)−18, which upregulates LAT2 expression in tumor cells and consequently enhances glutamine and leucine uptake- and mTOR activity-dependent CD47 expression for tumor immune envision.

## Results

### Osteosarcoma chemotherapy induces CD47 upregulation, which is associated with poor prognosis for patients

Immune checkpoint expression often changes following therapy[8,26–28]. To determine whether chemotherapy alters the expression of CD47, we performed immunohistochemistry (IHC) analysis to assess CD47 protein levels in paired specimens from 81 osteosarcoma patients before and after chemotherapy. As shown in Fig. 1a, the expression of CD47 (H-score) was significantly increased in the post-chemotherapy specimens compared to the paired pre-chemotherapy counterparts, suggesting that chemotherapy promotes CD47 expression in osteosarcoma. We next treated mice-bearing tumors derived from human osteosarcoma (HOS) with cisplatin, ifosfamide, methotrexate, or doxorubicin, the four most often used chemotherapeutic drugs for osteosarcoma, and detected obvious upregulation of mRNA and protein levels of CD47 in the tumors from cisplatin- or doxorubicin-treated mice (Fig. 1b and Supplementary Fig. 1a). This finding was validated by examination of CD47 expression on green fluorescent protein (GFP)-labeled HOS cells derived from tumors treated with these drugs by flow cytometry (Fig. 1c). These data suggested that

chemotherapy containing doxorubicin or cisplatin upregulates CD47 expression in osteosarcoma cells.

We next evaluated the outcomes of the chemotherapy based on the expression of CD47 in osteosarcoma patients. Consistent with previous studies[19,20], patients with high CD47 expression (CD47$^{high}$) pre-chemotherapy showed a shorter overall survival (OS) and disease-free survival (DFS) than those with low CD47 expression (CD47$^{low}$) (Fig. 1d and Supplementary Fig. 1b). Notably, patients with high upregulation of CD47 expression (ΔCD47$^{high}$) in response to chemotherapy also showed a shorter overall survival (OS) and disease-free survival (DFS) than that of patients with low CD47 upregulation (ΔCD47$^{low}$) (Fig. 1e and Supplementary Fig. 1c). We further stratified all these patients into four groups according to their CD47 expression pre-chemotherapy and upregulation of CD47 expression in response to chemotherapy, and found that patients with high levels of both CD47 expression pre-chemotherapy and CD47 upregulation (CD47$^{high}$ ΔCD47$^{high}$) had shortest survival among all these groups of patients (Fig. 1f and Supplementary Fig. 1d). These results strongly suggested that osteosarcoma chemotherapy induces CD47 upregulation in tumor cells, which is associated with poor prognosis for patients.

### Doxorubicin-induced CD47 upregulation in osteosarcoma is dependent on the activation of tumor-associated macrophages

To determine the mechanism underlying chemotherapy-induced CD47 upregulation in osteosarcoma, we treated different osteosarcoma cell lines with doxorubicin using a wide range of treatment time and doses, which surprisedly failed to obviously alter CD47 expression in the osteosarcoma cancer cells (Supplementary Fig. 2a–h), suggesting that doxorubicin-induced CD47 upregulation in osteosarcoma cells is tumor microenvironment-dependent.

The immune cells within the tumor microenvironment play important roles in tumorigenesis and cancer cell response to therapy[5,12]. In addition, macrophages are the most abundant immune cells in osteosarcoma[11,29,30] and are typically polarized into classically activated tumor-inhibiting macrophages (M1) and alternatively activated tumor-promoting macrophages (M2), with opposing effects on tumor cell viability[31]. Flow cytometry analysis showed that doxorubicin treatment did not change the percentages of myeloid-derived suppressor cells, dendritic cells, nature killer (NK) cells, B cells, or overall macrophages within osteosarcoma tumors (Supplementary Fig. 2i–m). However, doxorubicin treatment upregulated M1 marker MHC-II, inducible nitric oxide synthase (iNOS) and CD86 (Fig. 2a–c), and downregulated M2 marker Arginase 1 and CD206 (Fig. 2d, e) on tumor-associated macrophages. Quantitative real-time PCR analyses confirmed that doxorubicin enhanced the expression of M1 marker *Nos2*, *Cd80* and *Cd86*, and reduced expression of M2 marker *Arg1* and *Cd206* without altering the general macrophage markers, such as *CD11b* or *F4/80* (Fig. 2f). These results suggested that doxorubicin treatment activates macrophages with M1-porality, which are generally considered to elicit primarily immune-promoting effects.

Gene Expression Omnibus (GEO) dataset analysis showed that *CD47* expression positively correlates with M1 marker *CD80* (Supplementary Fig. 2n) and M1 abundance in tumor-infiltrating immune cells (Supplementary Fig. 2o). In the Therapeutically Applicable Research to Generate Effective Treatments (TARGET) osteosarcoma cohort, patients with higher *CD47* expression also showed higher M1 abundance (Supplementary Fig. 2p). To determine whether doxorubicin-activated macrophages are required for CD47 upregulation, we depleted macrophages in mice using clodronate liposomes. Successful depletion of macrophages in the tumor was confirmed by IHC staining for a macrophage marker F4/80 and flow cytometry analysis of CD11b$^+$ F4/80$^+$ cells (Fig. 2g and Supplementary Fig. 2q). Notably, clodronate liposome treatment reduced both basal CD47 expression and doxorubicin-induced CD47 upregulation in osteosarcoma (Fig. 2g and Supplementary Fig. 2r).

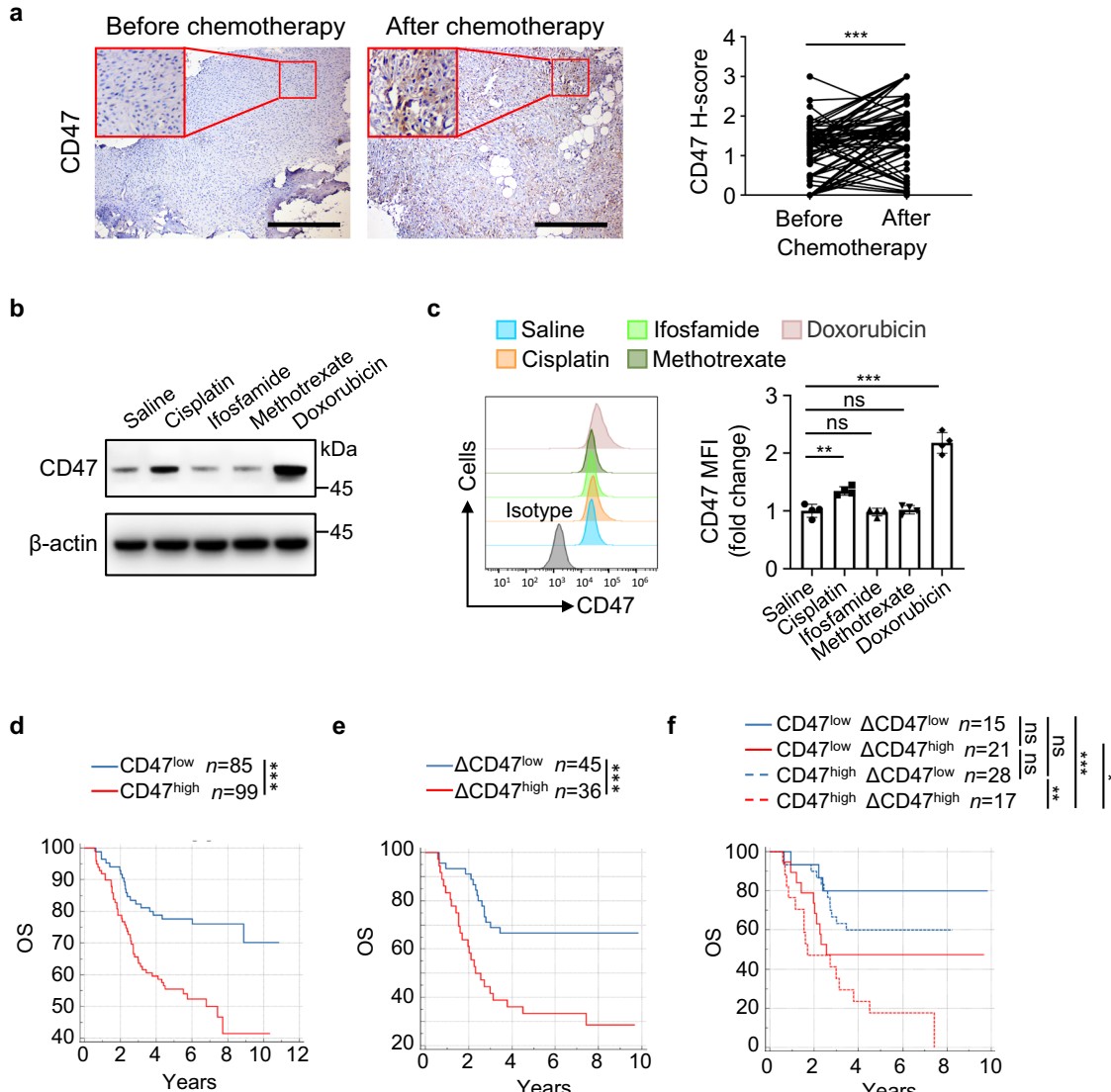

**Fig. 1 | Osteosarcoma chemotherapy induces CD47 upregulation, which is associated with poor prognosis for patients. a** Immunohistochemistry analysis of CD47 expression (CD47 H-score) in matched osteosarcoma specimens from the same patient before and after chemotherapy ($n = 81$ patients). Scale bar, 400 μm. **b**, **c** Cisplatin or doxorubicin was injected intravenously every 2 days, and ifosfamide or methotrexate was injected intravenously every 4 days, after GFP⁺ HOS tumor grew for 14 days. **b** Western blot analysis of CD47 expression in HOS tumors on day 22. **c** Flow cytometric analysis of CD47 protein expression in GFP⁺ HOS cells from tumors on day 22 (left; $n = 4$ mice per group). The anti-CD47 median fluorescence intensity (MFI) was determined (right). **d** Kaplan–Meier curves of overall survival (OS) of osteosarcoma patients stratified by CD47 expression in pre-chemotherapy specimens. High CD47 expression (CD47ʰⁱᵍʰ): CD47 H-score > 0.9; low CD47 expression (CD47ˡᵒʷ): CD47 H-score ≤ 0.9. **e**, **f** ΔCD47 H-score for each patient was obtained from subtracting CD47 H-score of pre-chemotherapy specimen from that of paired post-chemotherapy specimen. **e** Kaplan–Meier curves of

overall survival (OS) of osteosarcoma patients stratified by ΔCD47 expression. High ΔCD47 expression (ΔCD47ʰⁱᵍʰ): ΔCD47 H-score > 0.8; low ΔCD47 expression (ΔCD47ˡᵒʷ): ΔCD47 H-score ≤ 0.8. **f** Kaplan–Meier curves of overall survival (OS) of osteosarcoma patients stratified by CD47 expression in pre-chemotherapy specimens and ΔCD47 expression. Low CD47 expression and low ΔCD47 expression (CD47ˡᵒʷ ΔCD47ˡᵒʷ): CD47 H-score ≤ 0.9 and ΔCD47 H-score ≤ 0.8; low CD47 expression and high ΔCD47 expression (CD47ˡᵒʷ ΔCD47ʰⁱᵍʰ): CD47 H-score ≤ 0.9 and ΔCD47 H-score > 0.8; high CD47 expression and low ΔCD47 expression (CD47ʰⁱᵍʰ ΔCD47ˡᵒʷ): CD47 H-score > 0.9 and ΔCD47 H-score ≤ 0.8; high CD47 expression and high ΔCD47 expression (CD47ʰⁱᵍʰ ΔCD47ʰⁱᵍʰ): CD47 H-score > 0.9 and ΔCD47 H-score > 0.8. Data are shown as the mean ± SD. ns, not significant. *$P < 0.05$, **$P < 0.01$, ***$P < 0.001$, paired two-tailed Student $t$ test (**a**), one-way ANOVA (**c**) or log-rank test (**d–f**). The experiment was performed three times with similar results (**b**, **c**). See the Source Data file for the exact $P$ values. Source data are provided as a Source Data file.

We cocultured tumor cells with macrophages either by seeding them into the same culture well to allow their cell-cell contact (direct coculture) or seeding macrophages into the upper insert of a transwell apparatus with tumor cells in the lower well plates (indirect coculture). Doxorubicin treatment elicited similar levels of CD47 upregulation in these two conditions, suggesting that doxorubicin-upregulated CD47 expression primarily results from secreted soluble factors by the macrophages (Supplementary Fig. 3a). To determine the mechanism underlying macrophage-dependent CD47

upregulation in tumor cells, we generated macrophage-conditioned media by following these procedures: HOS cells were treated with DMSO or doxorubicin for 24 h, and then refreshed with serum-free medium for another 24 h to produce DMSO-treated tumor-conditioned medium (T-CMᴰᴹˢᴼ) or doxorubicin-treated tumor-conditioned medium (T-CMᴰᴼˣ). Next, THP-1-cell-derived macrophages were treated with the tumor-conditioned medium in the presence of DMSO or doxorubicin for 24 h, and then refreshed with serum-free medium for another 24 h to produce four kinds of macrophage-

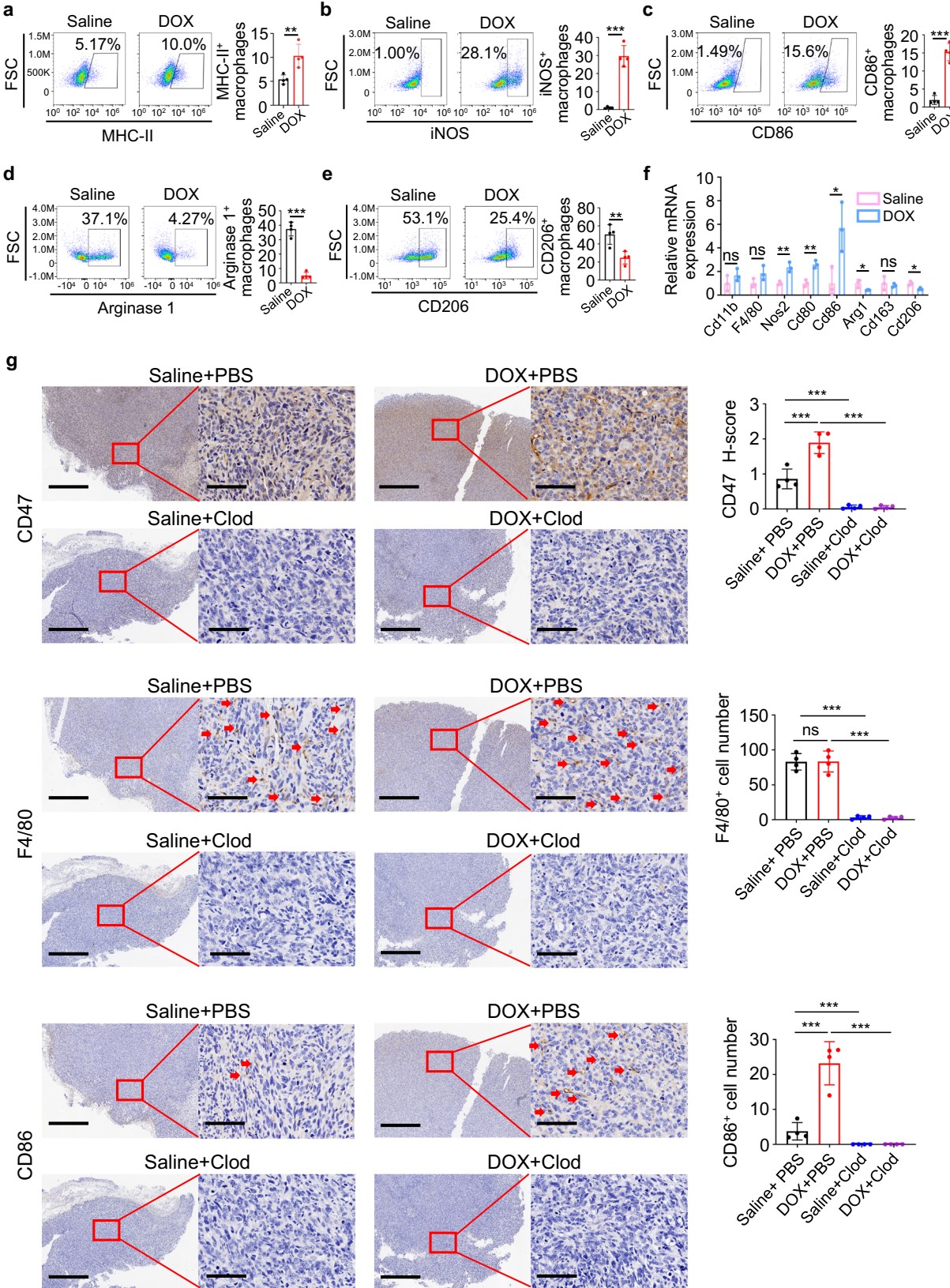

conditioned media (M-CM) (Fig. 3a). We only detected an obvious induction of CD47 expression in HOS, SJSA-1 and U-2 OS cells after incubation with conditioned medium of macrophage treated with doxorubicin in the presence of doxorubicin-treated tumor-conditioned medium (M-CM^DOX/DOX) (Fig. 3b, c and Supplementary Fig. 3b, c). Of note, the CD47 upregulation was accompanied by inhibited

macrophage phagocytosis of HOS and SJSA-1 cells (Fig. 3d, e), and this inhibition was abrogated by incubation with an anti-CD47 antibody, which remarkably enhanced the phagocytosis of HOS and SJSA-1 cells (Fig. 3d, e). These results suggested that doxorubicin combined with factors released by doxorubicin-treated osteosarcoma cells activate macrophages, which in turn secret soluble

**Fig. 2 | Doxorubicin-induced CD47 upregulation in osteosarcoma is dependent on the activation of tumor-associated macrophages. a–e** Flow cytometric analysis of MHC-II (**a**), iNOS (**b**), CD86 (**c**), Arginase 1 (**d**), and CD206 (**e**) expression in CD45$^+$ CD11b$^+$ F4/80$^+$ macrophages in HOS tumors on day 22 harvested from mice that received intravenous injection of doxorubicin (DOX) every 2 days ($n = 4$ mice per group). **f** Quantitative real-time PCR analysis of the indicated gene expression in HOS tumors treated as described in (**a**–**e**) on day 22 ($n = 3$ mice per group). **g** Immunohistochemical staining with antibodies as indicated in serial sections of HOS tumors treated with saline or doxorubicin (DOX), in the presence of clodronate (Clod) liposome or phosphate Buffer Saline (PBS) liposome on day 22. Scale bar, 800 μm and 80 μm, respectively. Representative IHC images are shown on the left. Quantitative analyses of IHC staining are shown on the right ($n = 4$ mice per group). Data are shown as the mean ± SD. ns not significant. *$P < 0.05$, **$P < 0.01$, ***$P < 0.001$, unpaired two-tailed Student $t$ test (**a**–**f**) or one-way ANOVA (**g**). The experiment was performed three times with similar results (**a**–**g**). See the Source Data file for the exact $P$ values. Source data are provided as a Source Data file.

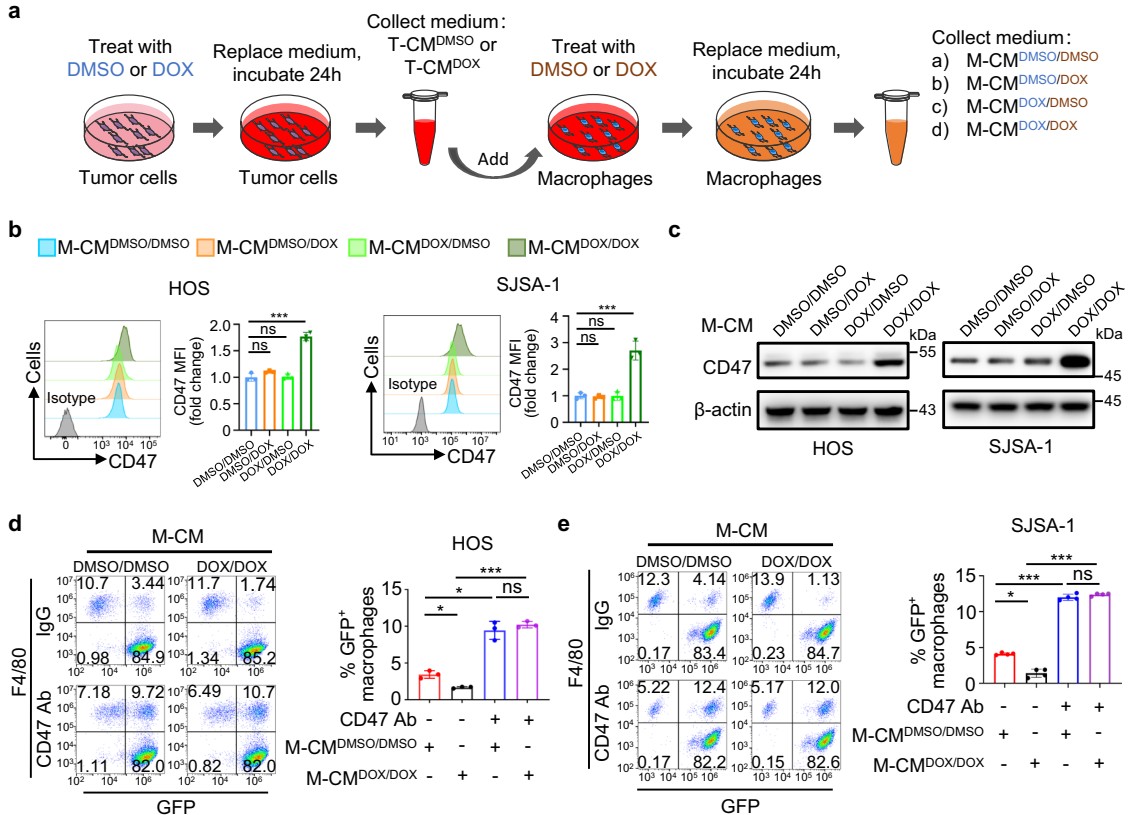

**Fig. 3 | Macrophage-secreted factors induced by doxorubicin promote CD47 expression in osteosarcoma cells for inhibition of macrophage phagocytosis. a** Schematic diagram of producing macrophage-conditioned media (M-CM). HOS cells were treated with DMSO or doxorubicin (DOX) to produce DMSO-treated tumor-conditioned medium (T-CM$^{DMSO}$) or DOX-treated tumor-conditioned medium (T-CM$^{DOX}$). THP-1-cell-derived macrophages were treated with the T-CM$^{DMSO}$ or T-CM$^{DOX}$ in the presence of DMSO or DOX to produce four kinds of M-CM: M-CM$^{DMSO/DMSO}$ was produced after treated with T-CM$^{DMSO}$ and DMSO; M-CM$^{DMSO/DOX}$ was produced after treated with T-CM$^{DMSO}$ and DOX; M-CM$^{DOX/DMSO}$ was produced after treated with T-CM$^{DOX}$ and DMSO; M-CM$^{DOX/DOX}$ was produced after treated with T-CM$^{DOX}$ and DOX. **b, c** HOS or SJSA-1 cells were treated with M-CM$^{DMSO/DMSO}$, M-CM$^{DMSO/DOX}$, M-CM$^{DOX/DMSO}$, or M-CM$^{DOX/DOX}$ for 24 h. **b** CD47 expression on the cell surface was determined by flow cytometry using isotype control or anti-CD47 antibody (left). The anti-CD47 median fluorescence intensity (MFI) was determined (right; $n = 3$ independent experiments). **c** Western blot analysis of CD47 expression in tumor cells. **d, e** Flow cytometry-based in vitro macrophage-mediated phagocytosis assay of HOS (**d**) or SJSA-1 (**e**) cells in the presence of IgG or anti-CD47 antibody (CD47 Ab) for 4 h ($n = 3$ independent experiments for HOS cells and $n = 4$ independent experiments for SJSA-1 cells). HOS or SJSA-1 cells were pretreated with M-CM$^{DMSO/DMSO}$ or M-CM$^{DOX/DOX}$ for 24 h. Macrophages were defined as F4/80$^+$ (labeled with PE) events, and tumor cells as GFP$^+$. F4/80$^+$, GFP$^+$ events represented macrophages that had phagocytosed tumor cells. Data are shown as the mean ± SD. ns not significant. *$P < 0.05$, ***$P < 0.001$, one-way ANOVA (**b, d, e**). The experiment was performed three times with similar results (**c**). See the Source Data file for the exact $P$ values. Source data are provided as a Source Data file.

factors to upregulate CD47 expression in tumor cells for inhibition of macrophage phagocytosis.

## Macrophage-secreted IL-18 induces CD47 expression in osteosarcoma cells

To identify the macrophage-secreted factors responsible for CD47 upregulation, we performed gene set enrichment analysis (GSEA) analyses with GEO data on osteosarcoma patients with high or low CD47 expression level. Genes in the Kyoto Encyclopedia of Genes and Genomes (KEGG) "cytokine–cytokine receptor interaction" gene set were significantly enriched in the high CD47 expression subgroup

(Supplementary Fig. 4a). Analysis of the cytokine amounts M-CM$^{DOX/DOX}$ revealed elevated amounts of IL-1ra and IL-18 (Fig. 4a). Treatment of HOS with IL-1ra and IL-18 showed that only IL-18 treatment induced the expression of CD47 (Fig. 4b), and IL-18 treatment induced upregulation of CD47 mRNA and protein in HOS and SJSA-1 cells (Fig. 4b and Supplementary Fig. 4b–d) was accompanied by an inhibition of phagocytosis of these cells, which was reversed by CD47 antibody (Fig. 4c and Supplementary Fig. 4e). In line with these findings in vitro, doxorubicin treatment induced upregulation of IL-18 in tumor tissue, which was abrogated by clodronate liposome-mediated depletion of macrophages (Fig. 4d, e). Intriguingly, the mRNA expression level of

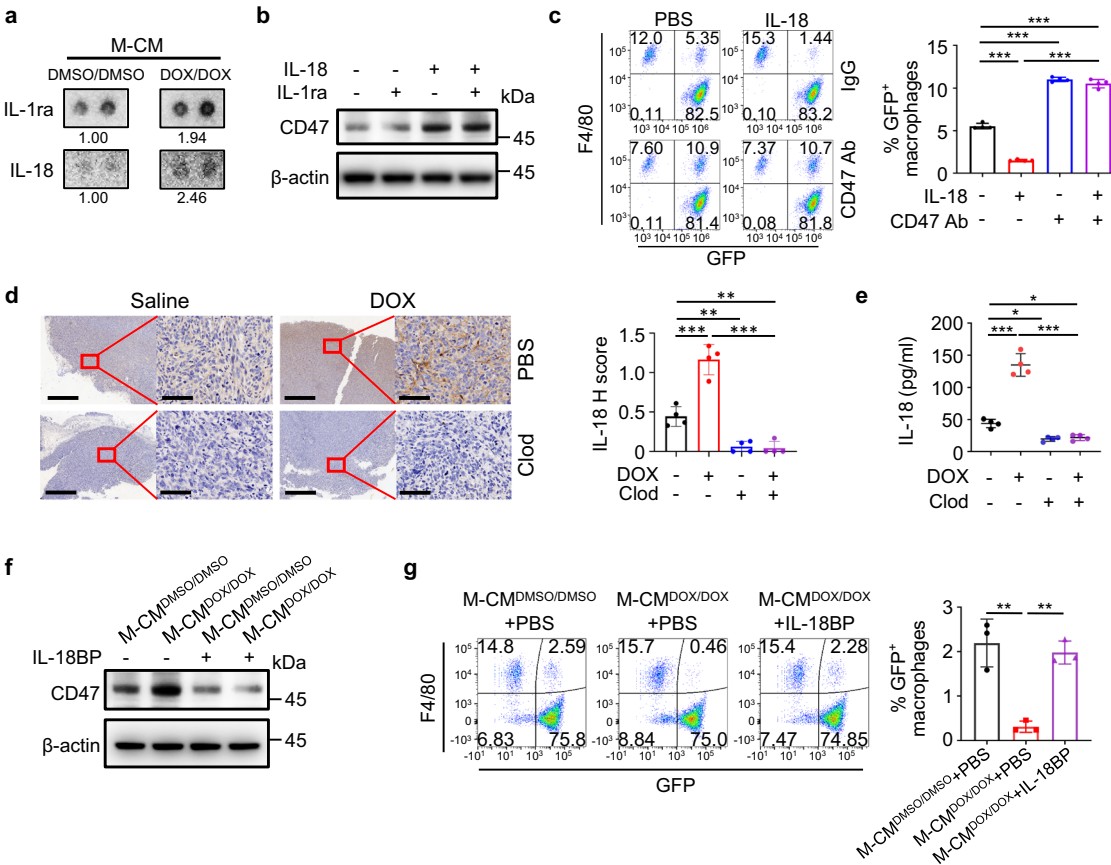

**Fig. 4 | Macrophage-secreted IL-18 induces CD47 expression in osteosarcoma cells. a** Analysis of cytokines in M-CM$^{DMSO/DMSO}$ or M-CM$^{DOX/DOX}$ by using a human cytokine array. Representative images for cytokines with elevated content in TAM-CM$^{DOX/DOX}$ are shown. Numbers indicate relative cytokine expression determined by mean spot pixel density. **b** Western blot analysis of CD47 protein in HOS cells serum-starved for 4 h followed by treatment with the indicated cytokines (40 ng/ml for each cytokine) for 24 h. **c** Flow cytometry-based in vitro macrophage-mediated phagocytosis assay of HOS cells in the presence of IgG or anti-CD47 antibody (CD47 Ab) for 4 h ($n = 4$ independent experiments). HOS cells were pretreated with PBS or IL-18 (40 ng/ml) for 24 h. Macrophages were defined as F4/80$^+$ (labeled with PE) events, and tumor cells as GFP$^+$. F4/80$^+$, GFP$^+$ events represented macrophages that had phagocytosed tumor cells. **d** Immunohistochemical analysis of IL-18 expression in serial sections of HOS tumors treated with saline or doxorubicin (DOX), in the presence of clodronate (Clod) liposome or phosphate buffer saline (PBS) liposome

on day 22 ($n = 4$ mice per group). Scale bar, 800 μm and 80 μm, respectively. **e** ELISA analysis of IL-18 concentrations in homogenates of HOS tumors treated with saline or doxorubicin (DOX), with clodronate (Clod) liposome or PBS liposome on day 22 ($n = 4$ mice per group). **f** Western blot analysis of HOS cells treated with M-CM$^{DMSO/DMSO}$ or M-CM$^{DOX/DOX}$ in the presence or absence of IL-18BP (2.5 μg/ml) for 24 h. **g** Flow cytometry-based in vitro macrophage phagocytosis assay. HOS cells were treated with M-CM$^{DMSO/DMSO}$, M-CM$^{DOX/DOX}$, or a combination of M-CM$^{DOX/DOX}$ and IL-18BP for 24 h ($n = 3$ independent experiments). Macrophages were defined as F4/80$^+$ (labeled with PE) events, and tumor cells as GFP$^+$ events. F4/80$^+$, GFP$^+$ events represented macrophages that had phagocytosed tumor cells. Data are shown as the mean ± SD. *$P < 0.05$, **$P < 0.01$, ***$P < 0.001$, one-way ANOVA (**c**–**e**, **g**). The experiment was performed three times with similar results (**a**, **b**, **d**–**f**). See the Source Data file for the exact $P$ values. Source data are provided as a Source Data file.

interleukin-18 receptor 1 (IL18R1) was significantly higher in osteosarcoma cell lines than in hFob1.19 normal osteoblast cell line (Supplementary Fig. 4f), implying that osteosarcoma cells are more sensitive to IL-18 than normal bone cells. These results suggested that doxorubicin-induced macrophage activation increases the secretion of IL-18 and subsequently enhances CD47 expression in osteosarcoma cells. Cisplatin, but not ifosfamide or methotrexate, could also increase the content of IL-18 in macrophage-conditioned media and tumors (Supplementary Fig. 5a, b), and the effects of these drugs on CD47 upregulation in tumor cells observed in vitro and in vivo (Fig. 1b, c and Supplementary Figs. 1a and 5c) were correlated with IL-18 levels induced by them.

We next neutralized IL-18 activity by adding its natural antagonist IL-18 binding protein (IL-18BP), which binds IL-18 with an affinity significantly higher than that of IL-18R[32–34], into M-CM$^{DOX/DOX}$. IL-18BP treatment ameliorated the macrophage-induced CD47 expression in HOS cells (Fig. 4f and Supplementary Fig. 6a) with the corresponding restoration of the phagocytosis of HOS cells by macrophages (Fig. 4g). Depletion of IL-18 expression in macrophage

(Supplementary Fig. 6b) showed similar effects with IL-18BP treatment on HOS and SJSA-1 tumor cells (Supplementary Fig. 6c–e). Analyses of the HOS-derived tumor tissues showed that doxorubicin increased macrophage activation with M1-polarity, IL-18 secretion, and CD47 expression (Fig. 5a, b and Supplementary Fig. 7a). Intra-tumoral IL-18BP injection in mice greatly decreased the basal CD47 expression and abrogated doxorubicin-induced CD47 upregulation (Fig. 5a, b and Supplementary Fig. 7a). Importantly, combined treatment with IL-18BP and doxorubicin significantly increased macrophage infiltration and activated macrophages with M1-polarity compared to single treatment (Fig. 5a, b) and resulted in significantly inhibition of tumor growth (Fig. 5c). Furthermore, knocking down of IL18R1 in HOS and SJSA-1 cells (Supplementary Fig. 7b) showed similar effects with IL-18BP treatment on suppressing CD47 expression (Fig. 5d, e and Supplementary Fig. 7c, d), enhancing macrophage infiltration and activation (Fig. 5d, e and Supplementary Fig. 7c, d) and inhibiting tumor growth (Fig. 5f and Supplementary Fig. 7e). These results suggested that doxorubicin-enhanced M1 macrophage activation increases IL-

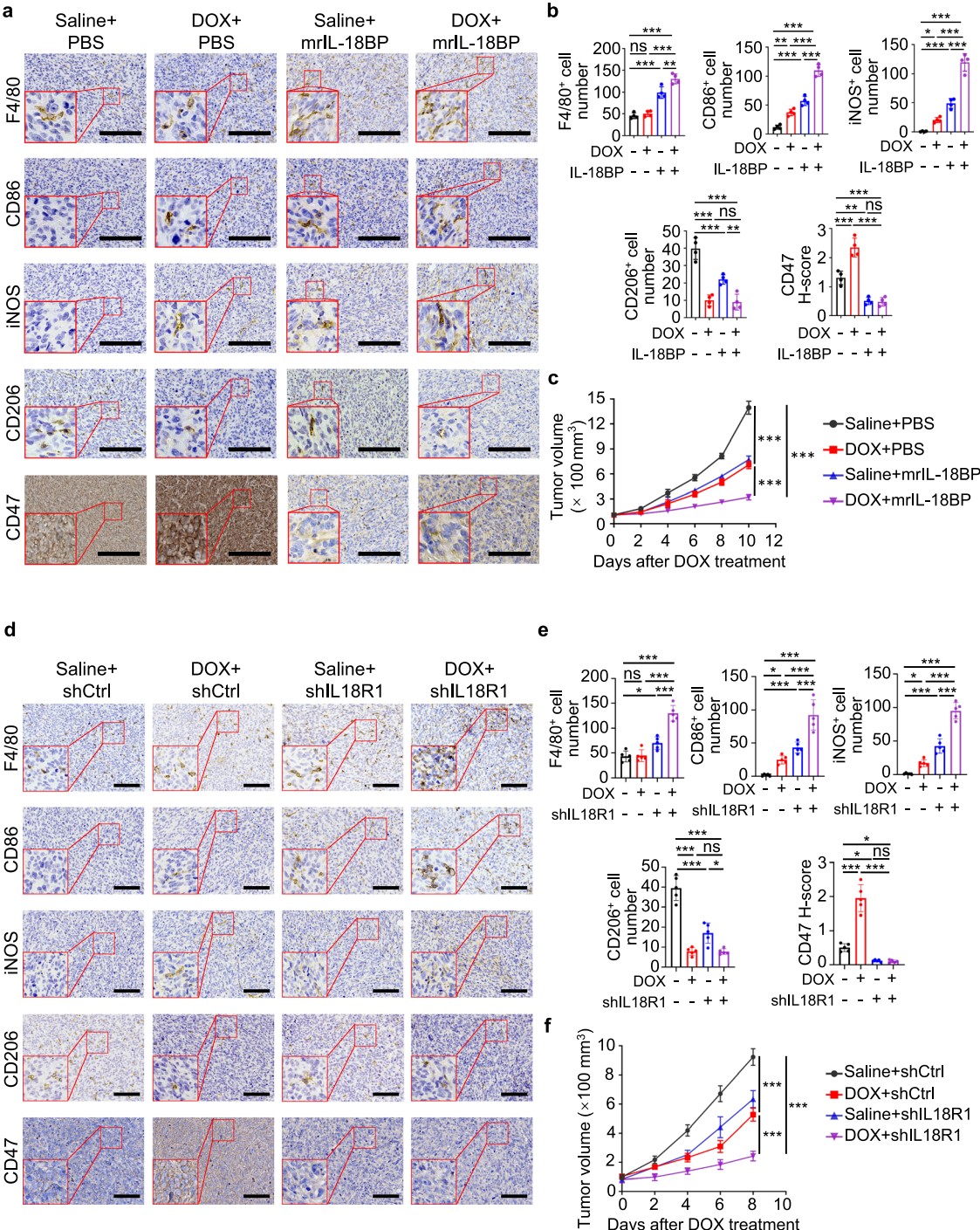

**Fig. 5 | IL-18 blockade sensitizes osteosarcoma to doxorubicin treatment.**
**a–c** HOS tumor-bearing mice were treated with saline or doxorubicin (DOX), with or without mouse recombinant (mr) IL-18BP. **a** Representative immunohisto-chemical images, showing F4/80, CD86, iNOS, CD206, and CD47 expression in serial sections of HOS tumors on day 22. Scale bar, 200 μm. **b** Statistical analyses of quantification of **a** (*n* = 4 mice per group). **c** Tumor growth of HOS cells in mice (*n* = 4 mice per group). Tumor volume was measured at the indicated time points. **d–f** Mice-bearing HOS tumors with or without IL18R1 depletion were treated with saline or doxorubicin (DOX) after tumors grew for 14 days. **d** Representative

immunohistochemical images, showing F4/80, CD86, iNOS, CD206, and CD47 expression in serial sections of HOS tumors on day 22. Scale bar, 300 μm.
**e** Statistical analyses of quantification of **d** (*n* = 5 mice per group). **f** Tumor growth of HOS cells in mice (*n* = 5 mice per group). Tumor volume was measured at the indicated time points. Data are shown as the mean ± SD. ns not significant. *P < 0.05, **P < 0.01, ***P < 0.001, one-way ANOVA (**b**, **e**) or two-way ANOVA (**c**, **f**). The experiment was performed three times with similar results (**a–f**). See the Source Data file for the exact *P* values. Source data are provided as a Source Data file.

18 secretion and CD47 expression and subsequent inhibition of macrophage phagocytosis.

To examine the clinical relevance of the finding, we analyzed GSE152048 single-cell RNA-seq data on 11 osteosarcoma tumors, which were collected after chemotherapy. In seven clusters of grouped cells

by unbiased clustering (Fig. 6a, Supplementary Fig. 8 and Supplementary Table 1), *IL18* was primarily expressed in the macrophages (Fig. 6b). In the TARGET osteosarcoma cohort, *IL18* expression was positively correlated with M1 marker *CD80* and *CD86* (Fig. 6c) and M1 abundance in tumor-infiltrating immune cells (Fig. 6d). In addition, we

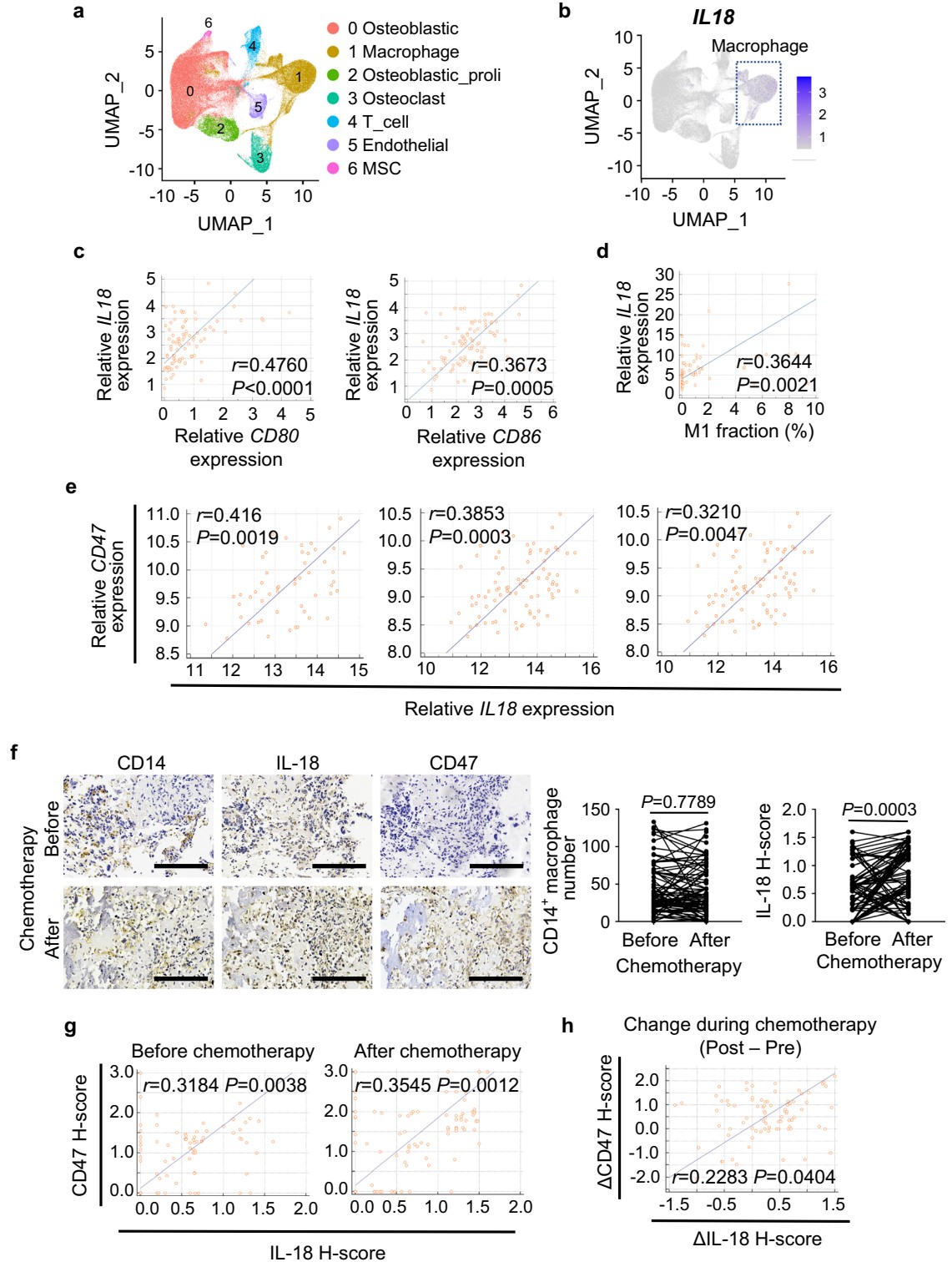

observed positive correlations between *CD47* and *IL18* in three different GEO cohorts of osteosarcoma patients (Fig. 6e). IHC staining of paired pre- and post-chemotherapy specimens from osteosarcoma patients revealed significant increase of IL-18 after chemotherapy without obvious change in total macrophage numbers (Fig. 6f). In addition, the expression of IL-18 and CD47 in osteosarcoma specimens before and after chemotherapy was correlated (Fig. 6g), and the changes of CD47 expression caused by chemotherapy were also correlated with the changes of IL-18 induced by chemotherapy (Fig. 6h). These results revealed the clinical relevance between macrophage-

secreted IL-18 levels and CD47 expression and the impact of chemotherapy on the enhanced CD47 expression via upregulated IL-18 secretion.

### Amino acid transporter LAT2-mediated amino acid uptake is necessary for IL-18-induced CD47 expression in osteosarcoma cells

GSEA analysis of TARGET osteosarcoma data showed that cell metabolism, including amino acid metabolism, was enriched in patients with high CD47 expression (Supplementary Fig. 9a). In

**Fig. 6 | IL-18 expression positively correlates with CD47 expression in osteosarcoma patients. a** Uniform Manifold Approximation and Projection (UMAP) plot showing clusters of seven main cell types (*n* = 11 osteosarcoma samples of patients). MSC, mesenchymal stem cell. **b** UMAP plot showing expression profiles of the *IL18* gene (*n* = 11 osteosarcoma samples of patients). **c** Scatter plots showing the correlation between *IL18* and M1 macrophage marker (*CD80* and *CD86*) gene expression in osteosarcoma patients from TARGET (*n* = 86 patients). Log₂ (TPM + 1) values were used for calculation and visualization. **d** Scatter plots showing the correlation between *IL18* gene expression and M1 macrophage fraction of immune cells in osteosarcoma patients from TARGET (*n* = 69 patients). M1 macrophage fraction was calculated by CIBERSORT. **e** Scatter plots showing the correlation between *IL18* and *CD47* gene expression in osteosarcoma patients from GSE21257 (left; *n* = 53 patients), GSE33382 (middle; *n* = 84 patients), and GSE30699 (right; *n* = 76

patients). **f** Representative immunohistochemical images showing CD14, IL-18, and CD47 expression in pre- and post-chemotherapy specimens from the same patient. Scale bar, 200 μm. Comparative analyses of immunohistochemistry-based quantification of CD14 and IL-18 expression are shown on the right. 81 pairs of matched specimens were analyzed (*n* = 81 patients). **g** Scatter plots showing the correlation between IL-18 and CD47 expression in osteosarcoma specimens before (left) or after (right) chemotherapy. **h** Scatter plots showing the correlation between changes of IL-18 and CD47 expression during chemotherapy. ΔIL-18/ΔCD47 H-score for each patient was obtained from subtracting IL-18/CD47 H-score of pre-chemotherapy specimen from that of paired post-chemotherapy specimen. Data are shown as the mean ± SD. Pearson correlation test (**c–e**, **g**, **h**) or paired two-tailed Student *t* test (**f**). Source data are provided as a Source Data file.

addition, it has been reported that IL-18 promotes leucine uptake by upregulating the expression of amino acid transporter LAT1 on NK cells[35]. IL-18 treatment significantly upregulated the mRNA expression levels of *LAT2*, but not *LAT1*, *LAT3*, or *LAT4*, in HOS cells (Fig. 7a) as well as LAT2 protein expression in these cells (Fig. 7b and Supplementary Fig. 9b). LAT2 associate with the 4F2 antigen heavy-chain (4F2hc) glycoprotein, forming a functional heterodimer[25], the mRNA expression of *4F2HC* was also increased by IL-18 (Fig. 7a). LAT2 depletion by expressing LAT2 shRNA remarkably decreased the basal (Fig. 7c and Supplementary Fig. 9c) and IL-18-induced CD47 expression in HOS cells (Fig. 7d). Consistently, treatment of HOS, SJSA-1 and U-2 OS cells with 2-amino-bicyclo-(2,2,1)-heptane-2-carboxylic acid (BCH), a specific inhibitor of LAT also diminished IL-18-induced CD47 expression (Fig. 7e and Supplementary Fig. 9d, e). In addition, LAT2 depletion (Fig. 7f and Supplementary Fig. 9f) or BCH treatment (Fig. 7f and Supplementary Fig. 9f) abrogated IL-18-suppressed macrophage phagocytosis of HOS cells. BCH treatment could also abrogate IL-18-suppressed macrophage phagocytosis of SJSA-1 cells (Supplementary Fig. 9g). These results indicated that LAT2 is indispensable for IL-18-upregulated CD47 expression and subsequent suppression of macrophage phagocytosis.

Similar to IL-18 treatment, M-CM^DOX/DOX also increased LAT2 expression on HOS cells, which was blocked by the depletion of IL-18 in macrophages (Supplementary Fig. 9h). IHC analysis of paired pre- and post-chemotherapy specimens from osteosarcoma patients revealed chemotherapy significantly increased LAT2 expression (Fig. 7g). Patients with high upregulation of LAT2 expression (ΔLAT2^high) showed worse overall survival and disease-free survival (Fig. 7h). In addition, the expression of LAT2 was positively correlated with expression of IL-18 (Supplementary Fig. 10a) as well as the expression of CD47 (Supplementary Fig. 10b) in osteosarcoma specimens before and after chemotherapy. The changes of LAT2 expression (ΔLAT2 H-score) increased by chemotherapy positively correlated with the changes of IL-18 (ΔIL-18 H-score) (Supplementary Fig. 10c), and the changes of CD47 expression (ΔCD47 H-score) increased by chemotherapy also positively correlated with the changes of LAT2 (Supplementary Fig. 10d). These results revealed the clinical relevance between LAT2 expression and IL-18/CD47 expression.

To determine the role of amino acids in IL-18-induced CD47 expression, we quantified intracellular amino acids in HOS cells and found that IL-18 treatment greatly increased the uptake of various amino acids, including glutamine (Gln) and leucine (Leu), and this increase was abrogated by LAT2 depletion (Fig. 7i). Notably, deprivation of amino acids in cell culture media, which did not affect IL-18-induced expression of LAT2, blocked IL-18-upregulated CD47 expression in HOS, SJSA-1, and U-2 OS cells (Fig. 7j and Supplementary Fig. 11a, b). These results indicated that LAT2-mediated amino acid uptake is necessary for IL-18-induced CD47 expression in osteosarcoma cells.

## IL-18-enhanced and LAT2-mediated uptake of glutamine and leucine promotes CD47 expression by activating the mTORC1/c-Myc axis

To identify which amino acid was necessary for IL-18-induced CD47 upregulation, we deprived the nine amino acids, whose concentrations in HOS cells were upregulated by IL-18 treatment, from the culture media. Deprivation of Gln or Leu, but not others, inhibited IL-18-induced CD47 expression (Fig. 8a and Supplementary Fig. 12a) and alleviated the IL-18-induced suppression of phagocytosis (Fig. 8b and Supplementary Fig. 12b). In addition, amino acid deprivation-suppressed CD47 expression in the absence and presence of IL-18 was abrogated by supplementation of Gln or Leu (Supplementary Fig. 12c). Deprivation of Gln or Leu also inhibited IL-18-induced CD47 expression in SJSA-1 and U-2 OS cells (Supplementary Fig. 12d, e). These data indicated that Gln and Leu are necessary and sufficient to induce IL-18-dependent CD47 expression and subsequent CD47-mediated inhibition of macrophage phagocytosis.

It has been reported that glutaminolysis and Leu could activate mammalian target of rapamycin complex 1 (mTORC1)[36,37] and that mTORC1 downstream effector c-Myc induces CD47 expression by binding directly to the CD47 promoter[38,39]. As expected, supplementation of Gln or Leu to amino acid-starved HOS cells increased phosphorylation of mTORC1 substrate p70 S6 Kinase (S6K), and expression levels of c-Myc in a dose-dependent manner (Supplementary Fig. 12f). Of note, IL-18 treatment also enhanced S6K phosphorylation and c-Myc expression, and this enhancement was reduced by deprivation of Gln or Leu from the media (Fig. 8c–e). Consistently, depletion of IL-18 in macrophages reduced M-CM^DOX/DOX-induced S6K phosphorylation and c-Myc expression in HOS cells (Supplementary Fig. 12g). All these results were indicating a Gln/Leu-activated mTORC1/c-Myc axis in mediating IL-18-induced CD47 expression. To further support this idea, treatment of HOS, SJSA-1, and U-2 OS cells with mTORC1 inhibitor rapamycin (Fig. 8f and Supplementary Fig. 13a) or c-Myc inhibitor 10058-F4 (Fig. 8g and Supplementary Fig. 13b) largely reduced IL-18-induced CD47 expression. In addition, rapamycin (Supplementary Fig. 13c, d) or 10058-F4 (Supplementary Fig. 13e, f) treatment abolished IL-18-suppressed phagocytosis of HOS and SJSA-1 cells by macrophages. Depletion of c-Myc also reduced IL-18-induced CD47 expression (Supplementary Fig. 13g) and phagocytosis suppression of HOS cells (Supplementary Fig. 13h). On the contrary, overexpression of c-Myc overcame the effects of Gln/Leu depletion to induce the expression of CD47 (Supplementary Fig. 14a) and decrease the phagocytosis of HOS cells (Supplementary Fig. 14b). These results suggested that IL-18-enhanced uptake of Gln and Leu activates mTORC1/c-Myc axis, which is required for IL-18-induced CD47 expression and inhibition of macrophage phagocytosis.

The role of LAT2 in the amino acid-dependent regulation of CD47 was further examined by LAT2 depletion, which blocked Gln and Leu-induced CD47 expression even in the presence of IL-18 (Fig. 8h). In addition, depletion (Fig. 8i) or inhibition (Fig. 8j) of LAT2 greatly reduced the basal and IL-18-enhanced mTORC1 activation and c-Myc

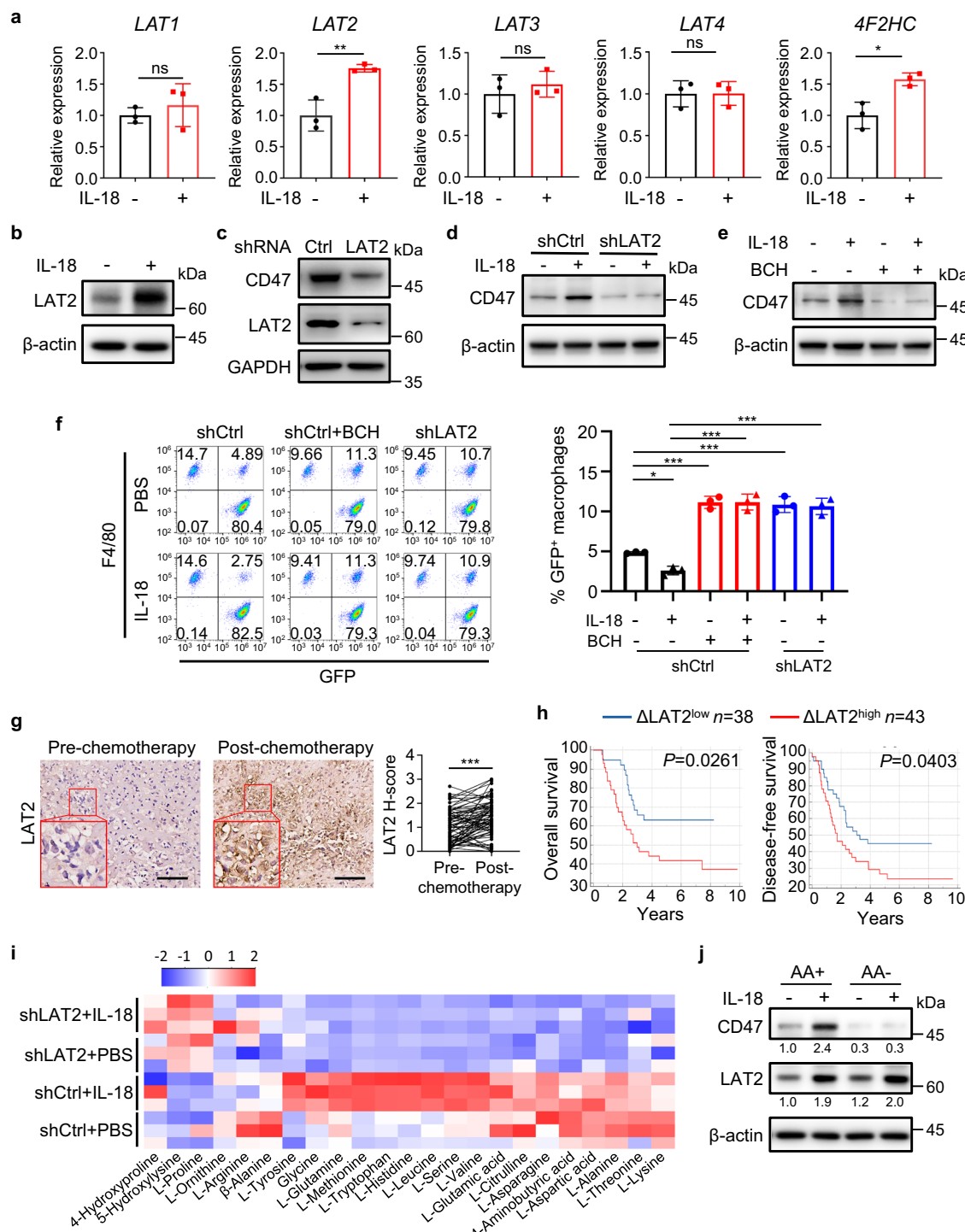

expression. Overexpression of c-Myc overcame the effects of LAT2 inhibition to induce the expression of CD47 (Supplementary Fig. 14c) and decrease the phagocytosis of HOS cells (Supplementary Fig. 14d). These results indicated that that LAT2-mediated uptake of Gln and Leu is indispensable for IL-18-induced mTORC1 activation and c-Myc expression for CD47 upregulation.

We then tested the antitumor effect of doxorubicin treatment in combination with MYC knockdown in HOS cells. c-Myc depletion greatly reduced basal and doxorubicin-induced CD47 expression in tumor tissues (Supplementary Fig. 15a). Compared with doxorubicin treatment or MYC depletion alone, combing MYC depletion with doxorubicin treatment greatly blunted tumor growth (Supplementary Fig. 15b) with increased infiltration of macrophages, and enhanced

activation of macrophages with M1 phenotype (Supplementary Fig. 15c). These data suggested c-Myc inhibition sensitizes osteosarcoma to doxorubicin treatment.

## LAT2 inhibition sensitizes osteosarcoma to doxorubicin treatment

To determine the role of LAT2-mediated amino acid uptake in doxorubicin-induced osteosarcoma immune evasion, we subcutaneously injected LAT2-depleted HOS cells into nude mice and treated the tumor-bearing mice with doxorubicin. Consistent with our observations from in vitro experiment (Fig. 7d), LAT2 depletion greatly reduced the basal and doxorubicin-induced CD47 expression in tumor tissues (Fig. 9a, b). Compared with doxorubicin treatment or LAT2

**Fig. 7 | Amino acid transporter LAT2-mediated amino acid uptake is necessary for IL-18-induced CD47 expression in osteosarcoma cells. a** Quantitative real-time PCR analysis of the indicated genes in HOS cells treated with IL-18 (40 ng/ml) or PBS for 24 h (*n* = 3 independent experiments). **b** Western blot analysis of LAT2 expression in HOS cells following IL-18 or PBS treatment for 24 h. **c** Western blot analysis of CD47 and LAT2 expression in shCtrl or shLAT2 HOS cells. **d** Western blot analysis of shCtrl or shLAT2 HOS cells treated with PBS or IL-18 for 24 h. **e** Western blot analysis of HOS cells treated with PBS or IL-18 in the presence or absence of BCH, for 24 h. **f** Flow cytometry-based analysis of macrophage phagocytosis of shCtrl HOS cells or shLAT2 HOS cells after treated as indicated. Macrophages were defined as F4/80⁺ (labeled with PE) events, and tumor cells as GFP⁺ events. F4/80⁺, GFP⁺ events represented macrophages that had phagocytosed tumor cells (*n* = 3 independent experiments). **g** Immunohistochemistry analysis of LAT2 expression (LAT2 H-score) in matched osteosarcoma specimens from the same patient before

and after chemotherapy (*n* = 81 patients). **h** Kaplan–Meier curves of overall survival (left) and disease-free survival (right) of osteosarcoma patients stratified by ΔLAT2 expression. High ΔLAT2 expression (ΔLAT2^high): ΔLAT2 H-score > 0.45; low ΔLAT2 expression (ΔLAT2^low): ΔLAT2 H-score < 0.45. ΔLAT2 H-score for each patient was obtained from subtracting LAT2 H-score of pre-chemotherapy specimen from that of paired post-chemotherapy specimen. **i** Heatmap showing the relative concentrations of amino acids in shCtrl or shLAT2 HOS cells stimulated with PBS or IL-18 (*n* = 3 independent experiments). **j** Western blot analysis of amino acid-starved HOS cells treated with IL-18 or PBS in amino acid-free (AA−) or amino acid-sufficient (AA+) medium for 21 h. Data are shown as the mean ± SD. ns, not significant. *$P < 0.05$, **$P < 0.01$, ***$P < 0.001$, unpaired two-tailed Student *t* test (**a**), one-way ANOVA (**f**), paired two-tailed Student *t* test (**g**) or log-rank test (**h**). The experiment was performed three times with similar results (**b–e**, **j**). See the Source Data file for the exact *P* values. Source data are provided as a Source Data file.

depletion alone, combination of LAT2 depletion with doxorubicin treatment greatly blunted tumor growth (Fig. 9c and Supplementary Fig. 16a) with increased infiltration of macrophages and enhanced activation of macrophages with M1 phenotype (Fig. 9a, b, d and Supplementary Fig. 16b). Notably, macrophage depletion using clodronate liposomes reduced the efficacy of the combination treatment (Fig. 9e).

In line with the results from LAT2 depletion, combination treatment with BCH sensitized the HOS- and SJSA-1-derived tumors to doxorubicin treatment (Fig. 10a and Supplementary Fig. 17a, b). As expected, BCH significantly downregulated both basal and doxorubicin-induced CD47 expression (Fig. 10b, c and Supplementary Fig. 17c). Combination treatment with BCH appeared to enhance the recruitment of macrophages, and promote macrophage activation towards M1 phenotype (Fig. 10b–d and Supplementary Fig. 17d, e). Consistently, macrophage depletion negated the efficacy of combination treatment (Fig. 10e and Supplementary Fig. 17f). These data suggested that inhibition of LAT2 sensitizes osteosarcoma to doxorubicin treatment by increasing macrophage phagocytosis.

Considering the dramatic drop of CD47 protein level after BCH treatment (Figs. 7e and 10b, c and Supplementary Fig. 17c), it is conceivable that the antitumor effect of BCH would be comparable with CD47 antibody treatment. As expected, animal studies showed that BCH and the CD47 antibody treatment exhibited similar levels of tumor growth inhibition (Supplementary Fig. 17g). In addition, a combination of doxorubicin and CD47 antibody treatment induced comparable anti-cancer effects in osteosarcoma to that induced by combine treatment of doxorubicin with BCH (Supplementary Fig. 17g).

## Discussion

Tumors are heterogeneous and complex tissues comprised of various cell types, among which the immune cells play dual roles in tumorigenesis. It has been well-established that the cytotoxic and phagocytic immune cells serve as the essential mechanisms to surveil and eliminate cancer cells[16,40]. Chemotherapeutic drugs can promote macrophage-mediated clearance of malignant cells by inducing pro-phagocytic "eat me" signal calreticulin on them[21,41], and skewed macrophages towards to antitumor phenotype, M1[7]. However, our results obtained from human patients and mice model demonstrated a side effect of the chemotherapy-activated immune response. Chemotherapy activates tumor-associated macrophages to secrete IL-18, which in turn induces LAT2 expression and LAT2-mediated Gln and Leu uptake in osteosarcoma cells. The enhanced Gln/Leu uptake stimulates the activation of mTORC1/c-Myc signaling to promote the expression of CD47 and leads to an inhibited macrophage phagocytosis of osteosarcoma cells (Fig. 10f). Thus, we revealed a regulatory mechanism underlying CD47 upregulation in osteosarcoma after chemotherapy, which resulted in an innate immune escape and was associated with poor prognosis in osteosarcoma patients. Our finding highlights the role of immune cells in acquired resistance to chemotherapy[27,42,43]. In triple-negative breast cancer cells, chemotherapy directly induces

hypoxia-inducible factor (HIF)-dependent enrichment of CD47[9]. However, in osteosarcoma cells, HIF-1α and HIF-2α expression levels were not obviously increased upon doxorubicin treatment (Supplementary Fig. 18a), suggesting that HIF-mediated signaling is not evident in inducing CD47 upregulation in osteosarcoma and mechanisms that regulate CD47 expression can be tumor type-dependent.

As a critical part of the CD47/SIRPα pathway, the expression of SIRPα in THP-1-derived macrophages and bone marrow-derived macrophages was not obviously upregulated by doxorubicin in the presence of doxorubicin-treated HOS conditioned medium (T-CM^DOX; Supplementary Fig. 18b). However, doxorubicin treatment of mice increased SIRPα expression in tumor-associated macrophages (Supplementary Fig. 18c), suggesting that secreted soluble factors from macrophage-surrounding cells in tumor microenvironment enhance SIRPα expression. These results also suggested that both enhanced expression of CD47 in tumor cells and upregulated SIRPα in tumor-associated macrophages plays a role in tumor immune evasion and that disrupting CD47 upregulation is effective for sensitizing doxorubicin treatment.

There are two emerging CD47-targeting strategies for cancer therapy, including interrupting the binding of CD47/SIRPα and downregulating CD47 expression at the transcriptional, translational, and post-translational levels[44]. Because of the high expression of CD47 in erythrocytes, the major side effect of clinical application of anti-CD47 antibodies and CD47-targeting SIRPα-Fc fusion protein is anemia due to CD47-mediated phagocytosis of erythrocytes[45–47]. Although designing novel antibodies and drug administration strategies reduces the side effects to a certain extent, whether antibodies can reach a balance between effectiveness and safety still need to be proved in clinic. RRx-001, inhibiting CD47 transcription, exhibits no obvious hematologic or systemic toxicity in clinic, suggesting that downregulating tumor CD47 expression by using small-molecule inhibitors may avoid hematological side effects[48,49]. Here, we identified a previously unreported therapeutic strategy to reduce CD47 expression by LAT2 inhibition and underscore the potential of LAT2 small-molecular inhibitors, such as BCH, in elimination of CD47-mediated tumor evasion.

High LAT2 expression is associated with poor overall survival in pancreatic cancer[50]. Furthermore, LAT2 regulates Gln-dependent mTOR activation to decrease chemotherapy sensitivity in pancreatic cancer[50]. Here, we found that macrophage-secreted IL-18 upregulated LAT2 in tumor cells, which enhanced the uptake of Gln and Leu to upregulate CD47 for inhibition of macrophage phagocytosis. Depletion or Inhibition of LAT2 in tumor cells abrogated IL-18-induced suppression of macrophage phagocytosis and improved antitumor immunity. This finding underscores the role of LAT2 in cellular communications in tumor microenvironment and mutual regulation between immune cells and tumor cells.

The uptake and metabolism of amino acids are aberrantly upregulated in many cancers to meet the demand for rapid growth and

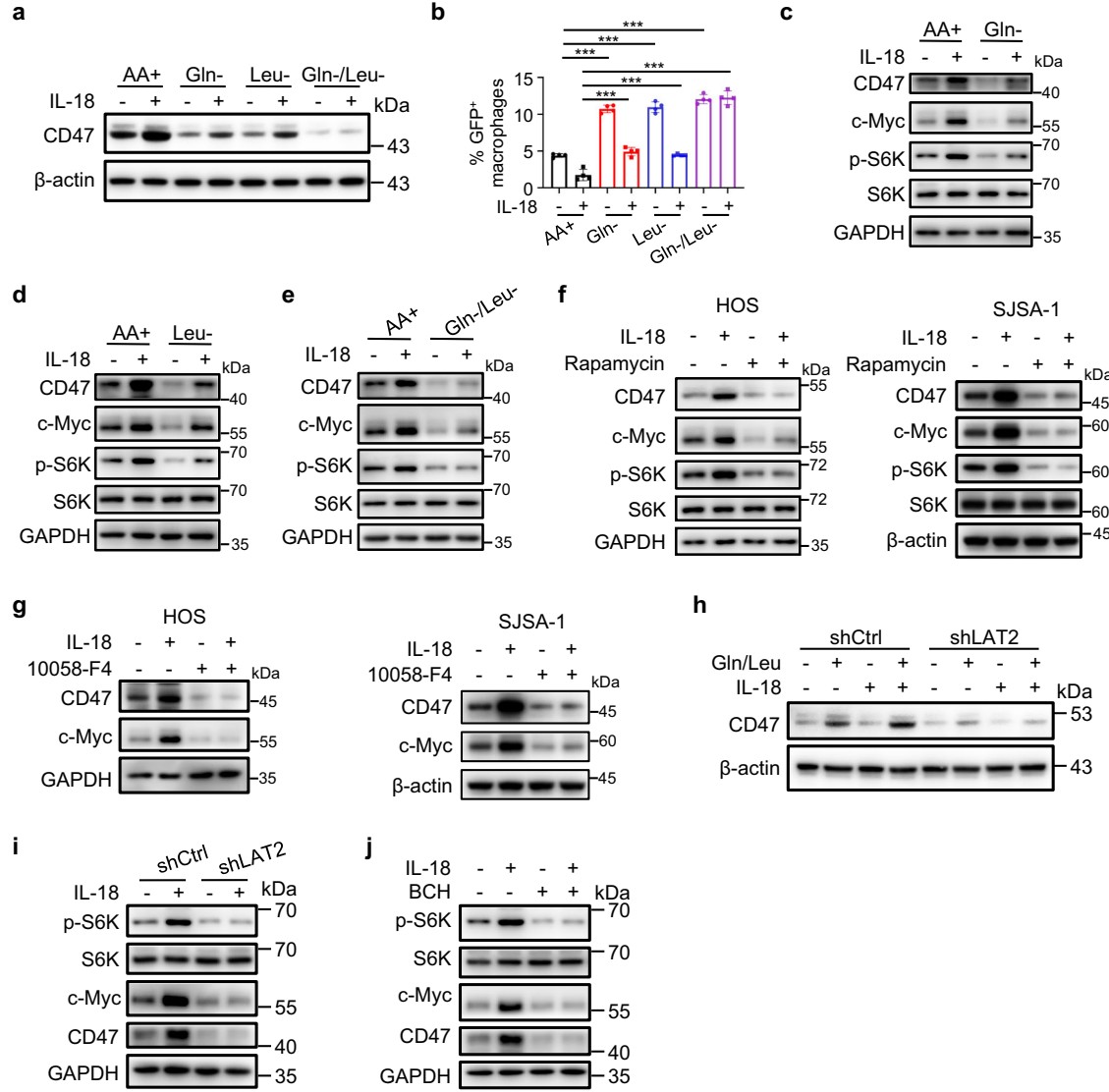

**Fig. 8 | IL-18-enhanced and LAT2-mediated uptake of glutamine and leucine promote CD47 expression by activating the mTORC1/c-Myc axis. a** Western blot analysis of HOS cells treated with PBS or IL-18 (40 ng/ml) in amino acid-sufficient medium (AA+) or medium lacking indicated amino acids for 24 h. Gln-, medium lacking glutamine; Leu-, medium lacking leucine; Gln-/Leu-, medium lacking glutamine and leucine. **b** Flow cytometry-based in vitro macrophage phagocytosis assay. HOS cells were treated with PBS or IL-18 (40 ng/ml) in amino acid-sufficient medium (AA+), medium deficient in leucine (Leu-), medium deficient in glutamine (Gln-), medium deficient in glutamine and leucine (Gln-/Leu-) for 24 h ($n = 4$ independent experiments). Macrophages were defined as F4/80+ (labeled with PE) events, and tumor cells as GFP+ events. F4/80+, GFP+ events represented macrophages that had phagocytosed tumor cells. **c–e** Western blot analysis of HOS cells treated with PBS or IL-18 (40 ng/ml) in amino acid-sufficient medium (AA+) or medium deficient in glutamine (Gln-, **c**), leucine (Leu-, **d**) or both glutamine and leucine (Gln-/Leu-, **e**) for 24 h. **f** Western blot analysis of HOS (left) or SJSA-1 (right) cells treated with PBS or IL-18 (40 ng/ml) in the presence or absence of rapamycin (25 nM) for 24 h. **g** Western blot analysis of HOS (left) or SJSA-1 (right) cells treated with PBS or IL-18 (40 ng/ml) in the presence or absence of 10058-F4 (100 μM) for 24 h. **h** Western blot analysis of amino acid-starved shCtrl or shLAT2 HOS cells, followed by the addition of indicated amino acids with or without IL-18 stimulation for 21 h. **i** Western blot analysis of shCtrl or shLAT2 HOS cells treated with PBS or IL-18 for 24 h. **j** Western blot analysis of HOS cells treated with PBS or IL-18 in the presence or absence of BCH, for 24 h. Data are shown as the mean ± SD. ***$P < 0.001$, one-way ANOVA (**b**). The experiment was performed three times with similar results (**a, c–j**). See the Source Data file for the exact $P$ values. Source data are provided as a Source Data file.

proliferation of tumor cells. Thus, targeting amino acid metabolism is becoming an attractive therapeutic strategy for cancer[23]. In addition, increasing evidence suggests an interconnection between tumor amino acid metabolism and cancer immunity[51]. Tumor cells impair T-cell functions and tumor immunity by outcompeting T cells for extracellular amino acids via high expression of selective transporters[51,52]. Gln blockade induces tumor metabolic reprograming, enhances tumor-specific immunity, and synergizes with T-cell immune checkpoint inhibitor treatment[53,54]. Here, we observed increased macrophage phagocytosis of osteosarcoma cells in response to

deprivation of Gln and Leu, revealing a role of amino acids in macrophage-mediated tumor immunity.

LAT2 expression in tumor-associated macrophages was much lower than tumor cells in osteosarcoma tissues (Supplementary Fig. 18d). The barely detectable LAT2 expression in tumor-associated macrophages suggested that the effect of the BCH on macrophages is minimal. Consistently, BCH treatment did not directly influence the macrophage phagocytosis of tumor cells in vitro (Supplementary Fig. 18e). These results suggested that BCH treatment does not obviously alter the function of macrophages directly.

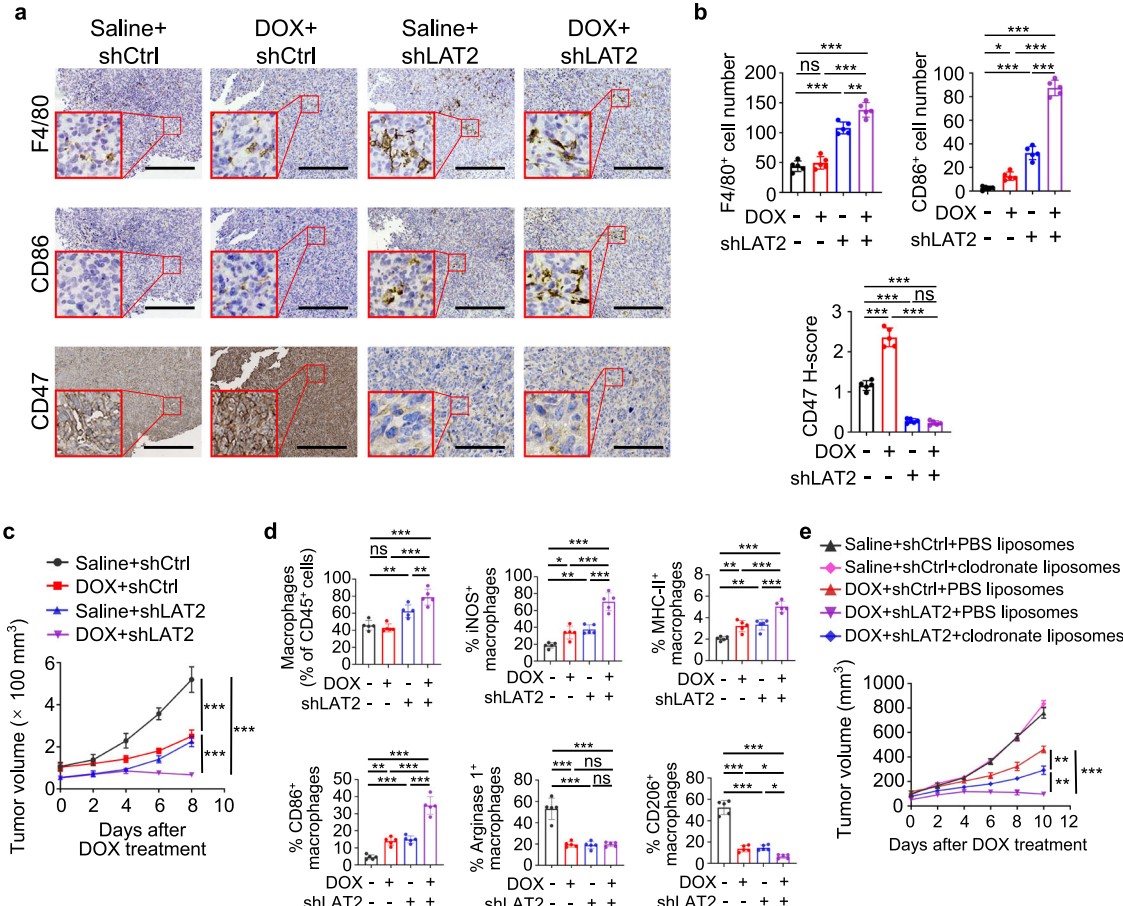

**Fig. 9 | LAT2 depletion sensitizes osteosarcoma to doxorubicin treatment.**
**a–d** Mice were treated with saline or doxorubicin (DOX) every other day after shCtrl or shLAT2 HOS tumors grew for 14 days. **a** Representative immunohistochemical images stained with F4/80, CD86, or CD47 antibodies in serial sections of tumors on day 22. Scale bar, 150 μm. **b** Statistical analyses of IHC quantification results for samples depicted in (**a**) (*n* = 5 mice per group). **c** Measurement of tumor growth in mice (*n* = 5 mice per group). Tumor volume was measured at the indicated time points. **d** Flow cytometric analysis of MHC-II, iNOS, CD86, Arginase 1, and CD206

expression in CD45⁺ CD11b⁺ F4/80⁺ macrophages in tumors on day 17 (*n* = 5 mice per group). **e** Volumes of tumors generated by shCtrl HOS cells or shLAT2 HOS cells after treatment as indicated (*n* = 5 mice per group). DOX, doxorubicin. Data are shown as the mean ± SD. ns not significant. *$P < 0.05$, **$P < 0.01$, ***$P < 0.001$, one-way ANOVA (**b**, **d**) or two-way ANOVA (**c**, **e**). The experiment was performed three times with similar results (**a–e**). See the Source Data file for the exact *P* values. Source data are provided as a Source Data file.

In summary, we unveiled a critical mechanism underlying chemotherapy-induced macrophage phagocytosis suppression, which is mediated by IL-18-induced and LAT2-dependent CD47 upregulation. This regulation confers cancer cell immune evasion. Inhibition of LAT2-mediated CD47 upregulation in osteosarcoma cells augments macrophage phagocytosis and sensitized tumor cells to chemotherapy. Thus, these findings conceptually advanced our understanding of the mechanism underlying chemoresistance and provided an attracted approach to sensitize cancer cells to chemotherapy.

## Methods

Our research complies with all relevant ethical regulations. Human studies were approved by the Human Research Ethics Committee of the Second Affiliated Hospital, Zhejiang University School of Medicine (SAHZU). Animal studies were approved by the Animal Ethics Committee of SAHZU (2018-009).

### Human osteosarcoma tissue samples

Osteosarcoma patients diagnosed between 2006 and 2014 at the Musculoskeletal Tumor Center of the Department of Orthopedics at SAHZU were included in this study for IHC analysis. These patients received chemotherapy regimens consisting of doxorubicin, cisplatin, and high-dose methotrexate, and some of them additionally received

ifosfamide. Formalin-fixed, paraffin-embedded osteosarcoma pre- and post-chemotherapy tissue blocks were collected. The detailed IHC procedures were described below. Areas in human tissue sections containing hemorrhage and obvious necrosis were excluded from pathology analysis. 184 patients were included in the final analyses, 103 of which had pre-chemotherapy specimens while 81 of which had paired pre- and post-chemotherapy specimens. The clinical characteristics of all the patients are summarized in Supplementary Table 2. Written informed consent was obtained from all patients. There was no participant compensation.

### Antibodies

Information on all antibodies used in this study is provided in Supplementary Table 3.

### Preparation of amino acid medium

To examine the direct effects of amino acid starvation and re-feed on tumor cells, amino acid-deficient and -sufficient media were prepared. Amino acid-free (AA−) medium was prepared using Dulbecco's Modified Eagle's Medium (DMEM) powder (D9800-27, US Biological Life Science), D-Glucose (4.5 g/L, the same concentration as in commercially-available DMEM) and sodium bicarbonate (3.7 g/L, the same concentration as in commercially-available DMEM), and

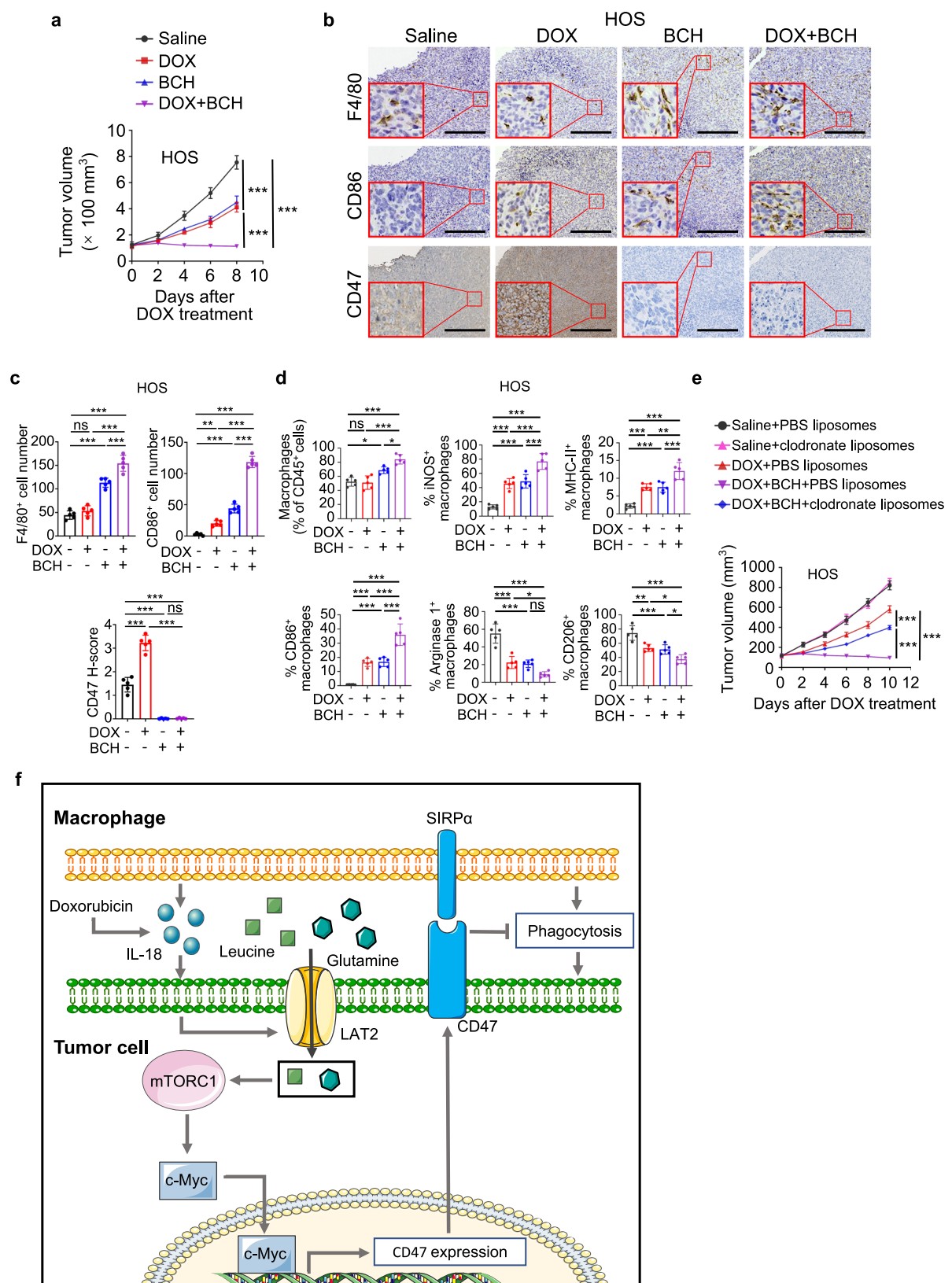

supplemented with 10% fetal bovine serum (FBS). Amino acid-sufficient (AA+) medium was prepared by adding appropriate amino acids to AA− medium to reach the same amino acid composition and content as commercially-available DMEM. Medium containing single amino acids (Gln or Leu) was prepared from AA− medium by adding individual amino acids at the same concentration as that present in AA+ medium. Media deficient for single amino acids (Gln or Leu) or a combination (Gln and Leu) thereof were prepared by adding amino acids to AA− medium, excluding the individual amino acids (with the same amino acid composition and content as in AA+ medium). Amino acids used in this study are shown in Supplementary Table 4.

**Fig. 10 | BCH sensitizes osteosarcoma to doxorubicin treatment. a–d** HOS tumor-bearing mice were treated with doxorubicin (DOX) along with BCH, 14 days after tumor incubation. **a** Measurement of tumor growth in mice ($n = 5$ mice per group). Tumor volume was measured at the indicated time points. **b** Representative immunohistochemical images stained with F4/80, CD86, or CD47 antibodies in serial sections of tumors on day 22. Scale bar, 150 μm. **c** Statistical analyses of IHC quantification results for samples depicted in **b** ($n = 5$ mice per group). **d** Flow cytometric analysis of MHC-II, iNOS, CD86, Arginase 1, and CD206 expression in CD45⁺ CD11b⁺ F4/80⁺ macrophages in tumors on day 17 ($n = 5$ mice per group).

**e** Tumor growth of HOS cells in mice treated as indicated ($n = 5$ mice per group). Doxorubicin (DOX) was intraperitoneally injected, and liposomes and BCH were intravenously injected. Tumor volume was measured at the indicated time points. **f** A graphical model showing chemotherapy-induced and macrophage-upregulated tumor cell CD47 expression. Data are shown as the mean ± SD. ns not significant. *$P < 0.05$, **$P < 0.01$, ***$P < 0.001$, two-way ANOVA (**a**, **e**) or one-way ANOVA (**c**, **d**). The experiment was performed three times with similar results (**a**–**e**). See the Source Data file for the exact $P$ values. Source data are provided as a Source Data file.

## Preparation of conditioned medium

HOS cells were treated with DMSO or 1 μM doxorubicin (S1208, Selleck Chemicals) for 24 h. Supernatants were then replaced with serum-free medium for another 24 h, to collect tumor-conditioned medium. THP-1 cells were treated with 320 nM phorbol-12-myristate-13-acetate (PMA; P1585, Sigma) for 6 h to acquire macrophages. THP-1-deprived macrophages serum-starved for 4 h prior to experiments were treated with DMSO or doxorubicin in the presence of the tumor-conditioned medium. After 24 h, culture supernatants were replaced with serum-free medium, to collect tumor-associated macrophage-conditioned medium for 24 h.

## Cell culture and stimulation

The human osteosarcoma cell lines, U-2 OS (SCSP-5030), MG63 (TCHu124), MNNG/HOS (TCHu167; abbreviated as HOS in this manuscript), the human osteoblast cell line, hFOB1.19 (GNHu14), the human monocytic cell line, THP-1 (TCHu 57) and the human embryonic kidney cell line, 293T (GNHu17) were obtained from Cell Collection of the Chinese Academy of Sciences. The human osteosarcoma cell line, 143B (CRL-8303), was obtained from American Type Culture Collection. The human osteosarcoma cell line, SJSA-1 (GDC0257), was obtained from China Center for Type Culture Collection. All cell lines were routinely examined for mycoplasma contamination and authenticated by DNA short tandem repeat genotyping. 293T, MG63, HOS, and 143B cells were cultured in DMEM containing 10% FBS in a 5% $CO_2$ atmosphere at 37 °C, unless otherwise stated, while U-2 OS, SJSA-1 and THP-1 cells were maintained in RPMI 1640 medium and hFOB1.19 cells were cultured in DMEM/F-12 medium.

For standard amino acid starvation, tumor cells were incubated in amino acid-free medium. Standard starvation time was 3 h, unless otherwise indicated. When required, cell supernatants were replaced with medium containing a single amino acid (Gln or Leu) for 21 h. For single amino acid or several amino acid starvation conditions, cells were incubated in medium deficient for a single amino acid (Gln or Leu) or a combination (Gln and Leu) thereof for 24 h. For human IL-18 (40 ng/ml; Biolegend), human IL-1α (40 ng/ml; PeproTech), human CXCL12 (40 ng/ml; PeproTech) or mouse IL-18 (40 ng/ml; PeproTech) treatment, cells were serum-starved for 4 h prior to cytokine stimulation at the indicated time points. Inhibitors, including 10058-F4 (100 μM; Selleck Chemicals), rapamycin (25 nM; Selleck Chemicals) and 2-amino-2-norbornanecarboxylic acid (BCH; 25 mM; Sigma-Aldrich) were added 1 h prior to cytokine stimulation. The macrophage-conditioned medium was used to stimulate tumor cells that were serum-starved for 4 h prior to conditioned medium exposure. IL-18BP (2.5 μg/ml; PeproTech) was added into macrophage-conditioned medium 1 h prior to conditioned medium stimulation to neutralize IL-18.

## Coculture of macrophages and osteosarcoma cells

Direct coculture: $1 \times 10^6$ THP-1 cells were seeded into six-well plates, polarized to macrophages with PMA as described above. After thoroughly washing, $3 \times 10^5$ HOS cells were added in the same culture well for indicated times.

Indirect coculture: $1 \times 10^6$ THP-1 cells were seeded into the upper insert of a six-well transwell apparatus (0.4 μm pore size, BD

Biosciences), polarized to macrophages with PMA as described above. After thoroughly washing, macrophages were cocultured with $3 \times 10^5$ HOS cells in six-well plates for indicated times.

## Proteome profiler human cytokine array assay

Cytokines and chemokines in M-CM^DMSO/DMSO or M-CM^DOX/DOX were measured according to the protocol for the Proteome Profiler Human Cytokine Array Kit (ARY005B, R&D Systems). Targeted proteins were visualized using an ultra-sensitive fluorescence/chemiluminescence imaging system Chemi Scope 6300 (CLiNX Science Instruments, Shanghai, China).

## Generation of stable cells using lentiviral infection

Lentivirus-containing shRNA used to knock down expression of human LAT2 (pHBLV-shLAT2) was purchased from Hanheng Biotechnology (Shanghai, China). The mature antisense sequence was 5′-CCAATG TCGCTTATGTCACTGCAAT-3′. Lentivirus-containing shRNA targeting human MYC (pGMLV-shMYC) was purchased from Genomeditech (Shanghai, China). The mature antisense sequence was 5′- GGAA-GAAATCGATGTTGTTTC-3′. IL18R1 shRNA plasmid (pGV248-shIL18R1) and IL18 shRNA plasmid (pGV248-shIL18) were purchased from Genechem (Shanghai, China). The mature antisense sequence was 5′-ACGTCTTCACAAGAGGAAT-3′ and 5′-TCCTGATAACATCAAGGAT-3′, respectively. pGV248-shIL18R1 or pGV248-shIL18 was co-transfected into 293T cells with lentiviral packaging plasmids to produce lentivirus expressing shRNA for IL18R1 or IL-18 using Hieff Trans liposomal transfection reagent (YEASEN; Shanghai, China), respectively. For LAT2, MYC, or IL18R1 knockdown in HOS cells, we infected HOS cells with lentivirus carrying pHBLV-shLAT2, pGMLV-shMYC, or pGV248-shIL18R1 with 7 μg/ml polybrene (H9268, Sigma), respectively. For IL-18 knockdown THP-1 cells, we infected THP-1 cells with lentivirus carrying pGV248-shIL18 with 8 μg/ml polybrene. After infection for 24 h, the medium was replaced with fresh medium, and 3 or 1 μg/ml puromycin (A1113802, Gibco) was used to select infected HOS cells or THP-1 cells, respectively. Lentivirus with noneffective shRNA (pHBLV-shCtrl, pGMLV-shCtrl, pGV248-shCtrl) provided by the manufacturer was used during the knockdown study as a control. The antisense sequence was 5′-TTCTCCGAACGTGTCACGT-3′.

The coding sequence for human MYC was cloned into the expression vector pGMLV (Genomeditech, Shanghai, China). The plasmids (MYC, or empty control) were transfected into 293T cells and lentivirus-containing medium supernatants were used to infect HOS cells with 7 μg/ml polybrene. Infected cells were selected using puromycin (3 μg/ml) methodology. We constructed GFP expression (GFP⁺) HOS cells using the same method.

## Measurement of intracellular amino acid content

Metabolites were extracted from HOS cells using extract solution (acetonitrile: methanol = 1:1, containing isotopically-labeled internal standard mixture). Standard solutions were prepared. An Agilent 1290 Infinity II series ultra-high-performance liquid chromatography (UHPLC) System (Agilent Technologies), equipped with a Waters ACQUITY UPLC BEH Amide column, was used to carry out UHPLC separation. Mass spectrometry was carried out using an Agilent 6460 triple quadrupole mass spectrometer (Agilent Technologies),

equipped with an AJS electrospray ionization (AJS-ESI) interface. Agilent MassHunter Work Station Software (B.08.00, Agilent Technologies) was employed for data acquisition and quantification of target amino acids. Final amino acid concentrations were normalized to total protein content (Supplementary Data 1 and 2).

## Quantitative real-time PCR

Total RNA was extracted using RNAiso reagent (Takara). Reverse transcription was achieved using a cDNA Synthesis Kit (Takara). mRNA levels were measured using gene-specific primers and SYBR Premix Ex Taq™ kits (Takara). Data were acquired using Real-Time PCR software (version 2.4). Expression results were normalized to β-actin levels. Primers were from PrimerBank[55–57] or bought from OriGene and can be found in Supplementary Table 5.

## Western blot

Total cellular protein extracts were obtained using RIPA lysis buffer (89901, Thermo Fisher Scientific) supplemented with protease inhibitor (78438, Thermo Fisher Scientific) and phosphatase inhibitor (78420, Thermo Fisher Scientific). Concentrations of extracted protein samples were tested using the BCA Protein Assay Kit (23227, Thermo Fisher Scientific). Equal quantities of protein extracts were separated by SDS-PAGE and then electroblotted on to polyvinylidene difluoride membranes (ISEQ00010, Merck Millipore), which were then blocked with 5% bovine serum albumin (BSA) for 1 h and incubated with primary antibodies, as indicated, at 4 °C overnight. The next day, membranes were washed and incubated with appropriate HRP-conjugated secondary antibodies for 2 h at 4 °C. Immunoreactive bands were detected using a Bio-Rad XRS chemiluminescence detection system (quantity one software v4.6.9) or a Amersham™ ImageQuant™ 800 Western Blot imaging system (ImageQuant 800 Control software v2.0.0).

## Enzyme-linked immunosorbent assay (ELISA)

Tumor tissue homogenates were assayed for mouse IL-18 using an ELISA assay kit (ab216165, Abcam), and macrophage-conditioned media were assayed for human IL-18 using an ELISA assay kit (ab215539, Abcam), according to the manufacturer's instructions. SoftMax Pro v7.1 software was used to acquire data.

## Flow cytometry and fluorescence-activated cell sorting (FACS)

Tumors were mechanically cut up using razor blades and then dissociated at 37 °C for 2 h in RPMI 1640 medium supplemented with collagenase type IV (2 mg/ml; Sigma), DNase (0.1 mg/ml; Sigma), hyaluronidase (0.1 mg/ml; Sigma), and BSA (2 mg/ml; Sigma). To acquire single cells, cell suspensions were passed through 100-μm filters. Cells grown adhered to culture plates were harvested using cell scrapers. For surface protein staining, cells were incubated in the dark with antibodies for 20 min at room temperature. For intracellular protein staining, cells were fixed with Fix/Perm solution (BD Biosciences), then washed with Perm/Wash buffer (BD Biosciences), and stained intracellularly for 20 min in the dark at room temperature. For the live/dead discrimination using flow cytometry analysis, cells were stained with Zombie Aqua™ Fixable Viability Kit (Biolegend) or the Zombie UV™ Fixable Viability Kit (Biolegend). Flow cytometry analyses were performed using Beckman Coulter CytoFLEX LX and CytExpert v2.4 software, and flow data were analyzed using FlowJo v10. For FACS, Beckman MoFlo Astrios EQs and Summit v6.3.1 software were used.

## Immunofluorescence (IF) and IHC

For IF staining, cells were plated in 24-well plates, washed with phosphate-buffered saline (PBS), fixed in 4% formaldehyde for 15 min at room temperature, and then permeabilized in 0.05% Triton X-100 in PBS for 10 min, followed by 10% BSA blocking for 20 min at room temperature. Cells were incubated overnight at 4 °C in blocking buffer containing antibody. Cells were washed with PBS and incubated with Alexa Fluor 555-labeled donkey anti-rabbit IgG for 2 h at room temperature. Cells were washed with PBS, and high-resolution images captured using microscopy (DMi8, Leica).

For immunohistochemical staining, formalin-fixed paraffin-embedded human osteosarcoma tissues or mouse xenograft samples were cut into 5-μm thick serial sections and subsequently incubated with antibodies overnight at 4 °C. The slides were then incubated with HRP-conjugated secondary antibodies, visualized using DAB, and counterstained with hematoxylin. Images were acquired using a microscope (Leica) and Leica Application Suite X v3.7.4 software.

Immunohistochemistry analysis was evaluated in five randomly selected fields of vision. Semi-quantitative histoscore (H-score) for CD47, LAT2, HIF-1α or HIF-2α was calculated by quantification of the percentage of positively stained cells using ImageJ (version 1.8.0) software. For IL-18, H-score was calculated by estimating the percentage of the positively stained area. Staining intensity was scored from 1 to 3. The following formula was used to calculate final H-score as follows: H-score = (low %) × 1 + (medium %) × 2 + (high %) × 3. CD14, F4/80, CD86, iNOS, or CD206 positive cells were counted based on the positive staining in each microscopic field. The optimal cut-off values of the H-scores from human osteosarcoma specimens for Kaplan–Meier analyses were determined by X-Tile software[58].

## Isolation of bone marrow-derived macrophages

Bone marrow cells were flushed out of 8-week-old female BALB/c mouse long bones with PBS. After isolation, cells were subjected to erythrocyte lysis (BD Bioscience) to remove the red blood cells and then cultured for 7 days in RPMI 1640 medium supplemented with 10% FBS, 1% penicillin/streptomycin, and 50 ng/ml CSF1 (Novoprotein).

## Phagocytosis assay

Bone marrow-derived macrophages were plated in a 24-well tissue-culture plate in a complete RPMI 1640 medium supplemented with 50 ng/ml CSF1 (Novoprotein), 24 h before the experiment. HOS cells were engineered to stably express GFP or labeled with CFSE (Biolegend) by suspending cells in PBS (5 μM working solution) for 20 min at 37 °C protected from light and washed twice with five times the original staining volume of media containing FBS before coculture. Macrophages were serum-starved for 2 h before adding GFP⁺ or CSFE⁺ HOS cells. HOS cells were treated as indicated in the legends. Macrophages and tumor cells were cocultured for 4 h at 37 °C. Anti-CD47 and control rat IgG antibody (Bio X cell) were added as indicated in the legends during the coculture. For flow cytometry-based phagocytosis assays, the ratio of macrophages to HOS cells was 1:4. Cells were harvested after coculture, stained with PE-conjugated F4/80 (Biolegend) to identify macrophages, and flow cytometry was conducted. Rates of phagocytosis were determined as the sum of GFP⁺ or CSFE⁺ macrophages as a percentage of total macrophages. For the immunofluorescence-based phagocytosis assay, macrophages were cocultured with an equal quantity of GFP⁺ HOS cells. Total cells were then incubated with PE-conjugated F4/80, as described above, and subsequently imaged using a fluorescence microscope (DMi8, Leica). Tumor phagocytosis was evaluated as the percentage of macrophages positive for phagocytized GFP⁺ cells.

## In vivo experiments

Mice were housed at 21 ± 2 °C with a 12 h dark/light cycle and fed free access to water and irradiated food. The housing environment was pathogen-free with humidity between 45 and 65%. All procedures using BALB/c nude mice were conducted adhering to the guidelines approved by the Animal Ethics Committee of SAHZU. Maximum tumor size (1500 mm³) permitted by the ethics committee was not exceeded. Five-week-old female BALB/c nude mice were each subcutaneously injected with 2 × 10⁶ HOS cells, GFP⁺ HOS cells, shCtrl HOS cells,

shLAT2 or shIL18R1 HOS cells suspended in 100 µl PBS on day 0. After 2 weeks, mice were divided into groups according to mean tumor volume for treatment. Doxorubicin (5 mg/kg; Selleck Chemicals) and cisplatin (6 mg/kg; Selleck Chemicals) were injected intravenously four times every 2 days, unless otherwise indicated. Ifosfamide (240 mg/kg; Selleck Chemicals) and methotrexate disodium (240 mg/kg, Selleck Chemicals) were administrated intravenously three times every 4 days. To reduce methotrexate disodium toxicity, folinic acid (50 mg/kg; Selleck Chemicals) were intraperitoneally administered 24 h after each dose of methotrexate disodium. For IL-18 neutralization assays, mouse IL-18BP (25 µg/kg; 122-BP, RD Systems) or normal saline was intratumorally injected for two cycles, with three days per cycle and a two-day interval between cycles. Mouse IL-18BP injection was initiated 1 day prior to first doxorubicin treatment. For macrophage depletion, 200 µL clodronate liposomes (Liposoma B.V.) or PBS liposomes (Liposoma B.V.) were administered through the caudal vein 3 days prior to tumor injection and every 4 days (seven times in total). BCH was administered intravenously at 200 mg/kg per mouse the day after the first doxorubicin treatment, then every 2 days, four times. Intraperitoneal injections of CD47 mAb (Bio X cell) at a dose of 10 mg/kg was initiated the day after the first doxorubicin treatment, then every 2 days for four times. 2 × 10[6] SJSA-1 cells, GFP[+] SJSA-1 cells, shCtrl SJSA-1 cells or shIL18R1 SJSA-1 cells were subcutaneously injected into BALB/c nude mice on day 0. Doxorubicin (5 mg/kg) was injected intraperitoneally five times every 3 days after tumor grew for 2 weeks. BCH (200 mg/kg) was injected intravenously after first doxorubicin treatment, and then every 2 days for 7 times. Clodronate liposomes were administered through the caudal vein 3 days prior to tumor injection and every 4 days for a total of nine times. Tumors were measured at indicated times and tumor volume was calculated using the formula: π/6 × length × width².

### GEO and TARGET data analysis
The results of this study were partly generated by analysis of data from TARGET and GEO datasets: GSE21257, GSE30699, GSE33382, and GSE152048. GSEA was performed using GSEA software[59]. CIBERSORT was used to estimate abundances of M1 macrophages in a mixed immune cell population[60]. For single-cell analysis, the main processing steps of data integration, cell-clustering, and annotation were performed using the R-based package, Seurat[61].

### Statistical analysis
Statistical analysis was performed using GraphPad Prism (version 7), Medcalc (version 19) or Statistical Product and Service Solutions (SPSS; version 23) statistical software. Values of $P < 0.05$ were considered statistically significant. All error bars in the data represent SD. Survival estimation was performed using the Kaplan–Meier method, and statistical differences were calculated using the log-rank test. Pearson correlation test was used for correlation analysis. Unpaired or paired two-tailed Student $t$ test was used to compare differences between the two groups. One-way ANOVA followed by the Tukey's multiple comparisons test was used to compare differences among multiple groups. Two-way ANOVA followed by the Tukey's multiple comparisons test was used to compare the difference between tumor growth curves.

### Reporting summary
Further information on research design is available in the Nature Research Reporting Summary linked to this article.

## Data availability
The publicly available data used in Fig. 6a, b and Supplementary Fig. 6a are available in the GEO database with accession number GSE152048. The publicly available data used in Fig. 6c, d and Supplementary Figs. 2p and 7a are generated by the TARGET [https://ocg.cancer.gov/programs/target] initiative, phs000468, with additional data available at https://portal.gdc.cancer.gov/projects. The publicly available data used in Fig. 6e and Supplementary Fig. 4a are available in the GEO database with accession numbers GSE21257, GSE30699, and GSE33382. The publicly available data used in Supplementary Fig. 2n, o are available in the GEO database with accession number GSE30699. The remaining data are available within the Article, Supplementary Information or Source Data file. Source data are provided with this paper.

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

## Acknowledgements

This study was supported by grants from the National Natural Science Foundation of China (81872173 to Z.Y.; 82072959 to Z.Y.), Natural

Science Foundation of Zhejiang Province (LD21H160002 to Z.Y.), Medical and Health Science and Technology Plan of Department of Health of Zhejiang Province (WKJ-ZJ-1821 to Z.Y.), and China Postdoctoral Science Foundation (2021M692792 to Z.W.). Figures 3a and 10f were modified from Servier Medical Art (http://smart.servier.com/), licensed under a Creative Common Attribution 3.0 Generic License (https://creativecommons.org/licenses/by/3.0/).

## Author contributions

Z.Y. and Zenan W. conceived and designed the study. Z.L. and S.L. provided critical scientific input. Zenan W., B.L., S.L., W.L., Zhan W., S.W., W.C., W.S., T.C., H.Z., E.Y., W.Z., H.M., X.C., and J.Z. performed the experiments. Zenan W. wrote the draft manuscript. Z.Y., Z.L., and S.L. revised the manuscript.

## Competing interests

The authors declare no competing interests.

## Additional information

**Correspondence and requests** for materials should be addressed to Zhaoming Ye.

