## [Peer Review File · Nature Communications]

Metabolic control of CD47 expression through LAT2-mediated amino acid uptake promotes tumor immune evasionREVIEWER COMMENTS

Reviewer #1 (Remarks to the Author): with expertise in macrophages, CD47

In this study by Wang et al, the authors showed enhanced CD47 expression in osteosarcoma tissues after chemotherapy treatment. Further investigation by the authors demonstrated a role of IL18, secreted by macrophages in response to doxorubicin, in inducing LAT2 expression for leucine and glutamine uptake which in turn activated mTORC1 and c-Myc for enhancing CD47 expression. Their findings revealed important immune evasion mechanisms developed by osteosarcoma during chemotherapy and identified the underlying mechanisms for CD47 upregulation. This study was well performed with proper controls.

Despite the enthusiasm, there is a lack of convincing evidence proving some of the key conclusions and/or connecting the findings from different experiments, which need to be adequately addressed by the authors (detailed in below).

Major concerns:

1. In Fig3, the role of IL18 in doxorubicin-induced CD47 upregulation was examined by IL18BP. However, whether and how IL18BP functions as an antagonist of IL18 was not established. If IL18BP functions through competing with endogenous IL18 for binding to IL18 receptors on osteosarcoma cells, a binding assay should be performed to demonstrate such effects. In addition, additional IL18 blocking approaches should be used to confirm its role – eg. knockdown of IL18 expression in macrophages. Moreover, *in vivo* experiments with osteosarcoma cells deficient in responding to IL18 (eg. IL18R knockdown osteosarcoma cells) should be performed to demonstrate doxorubicin can not induce CD47 upregulation in such cells.

2. In Fig5, the authors claimed the activation of mTORC1 led to subsequent c-Myc-mediated transcription of CD47. However, their data only suggested IL18 induced c-myc expression and S6K phosphorylation which were abolished by rapamycin. There were not sufficient data showing enhanced CD47 expression was regulated by mTORC1 and c-Myc. This can be addressed by examining the effects of a knockdown of S6K and/or c-Myc on LAT2-mediated CD47 expression.

3. The clinical significance of combining doxorubicin and LAT2 inhibitors is unclear. As shown in Fig2F, although doxorubicin induced resistance to phagocytosis by increasing CD47 expression, this can be overcome by CD47 antibody which strongly induced phagocytosis of both untreated and doxorubicin-treated cells. Therefore, one would predict the upregulation of CD47 by doxorubicin would not compromise the efficacy of CD47 antibody, and a combination of doxorubicin and CD47 antibody treatment may induce stronger anti-cancer effects in osteosarcoma than that by doxorubicin+BCH. These should be examined with the *in vivo* models.

4. In the phagocytosis assays shown in Fig4F, Fig5G, etc., it's not convincing the changes of phagocytosis were due to up- or down-regulation of CD47. A control group of osteosarcoma cells (CD47 knockdown or anti-CD47 Fab treatment to block CD47) should be included to demonstrate the effects of IL18 and/or BCH/rapamycin on phagocytosis are abolished when CD47 is blocked.

Minor issues:

1. Fig1, the experimental details of the *in vivo* experiments should be described;
2. Fig2C, Fig3E, Fig6A, and FigS2G, the expression levels of the markers examined should be quantified.
3. Fig2 and Fig3, the conditioned medium generated from THP1 cell line should not be termed TAM conditioned medium.

Reviewer #2 (Remarks to the Author): with expertise in cancer immunometabolism

In this manuscript by Wang et al., the authors present evidence that LAT2-mediated amino acid uptake controls doxorubicin-induced macrophage evasion by osteosarcoma cells. Using the human HOS cell line as a primary model, the authors demonstrate that doxorubicin treatment enhances the expression of the immune checkpoint protein CD47 on tumor cells in vivo and propose a macrophage-dependent mechanism. The authors extend this observation to a small osteosarcoma cohort of paired pre- and post-chemotherapy specimens, and found that patients with higher CD47 expression post-chemotherapy had shorter overall and disease-free survival. Using tumor-associated macrophage (TAM) conditioned media, the authors identified that doxorubicin promotes the secretion of IL-18 by macrophages. Supplementing tumor cells with IL-18 was sufficient to induce CD47 expression. In contrast, antagonizing IL-18 abrogated the increase in CD47 expression and enhanced phagocytosis of tumor cells by macrophages. The authors subsequently showed that IL-18 promotes the expression of the amino acid transporter LAT2 in osteosarcoma cells, as well as uptake of the amino acids Gln and Leu. Genetic depletion of LAT2, pharmacological inhibition of the LAT proteins, or depletion of Gln/Leu in tumor cells diminished IL-18-induced expression of CD47 and enhanced phagocytosis by macrophages. The authors further showed that supplementing amino acid-starved tumor cells with Gln or Leu increased mTORC1 activity and c-Myc expression, which were abrogated by LAT2 depletion or inhibition. Treatment of tumor cells with rapamycin or a c-Myc inhibitor phenocopied LAT2 inhibition and diminished IL-18-induced CD47 expression. Finally, using a series of HOS xenograft experiments, the authors demonstrated that LAT2 depletion or inhibition enhances the efficacy of doxorubicin in vivo, which they claim is associated with increased M1 macrophage infiltration into the tumor.

The manuscript is clearly written, and the main conclusions have important implications for tumor immunology and immunotherapy. Nonetheless, there are a few issues that need to be addressed to fully support the authors' conclusions before this manuscript can be considered for publication.

Major comments:

1. In the Abstract and throughout the manuscript, the authors claim that chemotherapy enhances the expression levels of CD47; however, only doxorubicin was evaluated in this study. Is the secretion of IL-18 by TAMs and the subsequent upregulation of CD47 by osteosarcoma cells specific to doxorubicin, or can other chemotherapies that are used to treat osteosarcoma elicit a similar response?
2. Throughout the manuscript, summarized flow cytometry data are presented (e.g., Figure 4F, 5B, 5G, S1B, etc.), but the corresponding flow cytometry plots are not shown. These plots should be provided as supplementary figures.
3. In Figure 2C, 3E, 6A, 6G, etc., the authors draw conclusions from qualitative IHC staining. Quantification of the staining (e.g., average number of positive cells per tumor area, pixel intensity, etc.) would significantly aid in data interpretation. For example, the authors use clodronate liposomes in the study to deplete macrophages and claim that this treatment reduces F4/80 immunostaining in tumors (Figure 2C); however, this conclusion is not supported by the small 'representative' images shown by the authors. Quantification of F4/80 and CD86 staining is required to support the authors' claims.
4. As an extension to #4, the authors claim that doxorubicin + LAT2 depletion (Figure 6A) or LAT2 inhibition (Figure 6G) enhances the recruitment of macrophages into the tumor. This conclusion is not supported by the IHC data. The authors should quantify the F4/80 and CD86 staining. In addition, the authors could quantify TAMs (total, M1, and M2) by flow cytometry, as the authors did in Figures 1 and S1, at an early time-point when tumor volumes are similar between treatment groups.
5. In Figure 5, the authors demonstrate that treatment of HOS cells with either rapamycin (Figure 5E) or 10058-F4 (Figure 5F) impairs IL-18-induced CD47 expression; however, only rapamycin was

further evaluated and shown to enhance phagocytosis of HOS cells by macrophages. Does inhibition of c-Myc with 10058-F4 also enhance phagocytosis?

6. In the Discussion (lines 331-334), the authors cite a study that reported HIF-dependent enrichment of CD47 in response to chemotherapy in triple-negative breast cancer cells and state that regulation of CD47 can be tumor type-dependent. Did the authors evaluate HIF expression in their osteosarcoma cells following doxorubicin treatment or evaluate common markers of hypoxia (e.g., pimonidazole) in vivo? Is this mechanism also evident in osteosarcoma cells?

7. In Supplementary Figure S2H, the authors demonstrate that mRNA expression of IL18R1 is significantly higher in osteosarcoma cell lines compared to a normal osteoblast cell line. Does doxorubicin treatment further increase the expression of IL18R1 in osteosarcoma cells?

8. A major limitation of the study is that the authors rely primarily on a single cell line (HOS cells). While other osteosarcoma lines are periodically assayed in the study, the authors should ensure that their main conclusions are supported by data from more than one cell line. For example, does LAT2 depletion or inhibition sensitize U-2 OS cells to doxorubicin in vivo?

Minor comments:

1. Many figure legends do not state whether the error bars represent the standard error of the mean (SEM) or standard deviation (SD). This should be made clear for each figure, where appropriate.

2. For the overlaid histograms in Figures 1D, 2A, and 2E, the data should be normalized to mode. Moreover, in Figure 2A, the quantification of MFI for CD206 is not appropriate, as two distinct populations are apparent. Reporting the percentage of CD206+ cells would be more appropriate.

3. What gating strategy was used for Figure 2A? What is the parent population of the CD86+ and CD206+ cells? This should be reported in the figure and/or figure legend.

4. What markers were used to distinguish each cell type in Supplementary Figure S1B? This should be reported in the figure and/or figure legend.

5. The authors use the word “synergistic” to describe the interaction between doxorubicin and BCH (line 307 and 373); however, the authors do not formally evaluate whether the interaction between doxorubicin and BCH is additive or synergistic. In the absence of this analysis, the authors should refrain from using the word “synergistic”.

Reviewer #3 (Remarks to the Author): with expertise in osteosarcoma, chemotherapy

The present manuscript describes the induction of immunosuppression under chemotherapy in the context of osteosarcoma. The Authors highlighted the key role of the tumor microenvironment, notably through induction of IL18 secretion by the macrophages. The activation of IL18 receptor at the surface of tumor cells led to an upregulation of LAT2, the subsequent modulation of the amino acid metabolism (mainly Leu and Gln), and the activation of mTOR signaling pathway leading to the ignition of c-myc targeted promoter of the CD47 gene. The upregulation of CD47 at the tumor cell surface triggers the inhibition of macrophage phagocytosis, and thus correlates with tumor progression and worse prognosis.

The Authors used various models: modified HOS cell lines in vitro and in vivo (CDX models), osteosarcoma tissue samples from patients, and bioinformatics database (GEO).

Complementary approaches were used to fully confirm their hypotheses. The manuscript is well written, and the conclusions are in line with the results. However, in order for this paper to be fully

suitable for publication, several minor points have to be considered.

---Major comments

The Authors indicate: "patients received chemotherapy regimens consisting of doxorubicin, cisplatin, and high-dose methotrexate, and some of them additionally received ifosfamide" (Line 398), but tested only doxorubicin for the IL18/CD47 induction. What about other conventional drugs onto macrophage activation and crosstalk with tumor cells?

Convincing results suggest that high CD47 levels correlate with poor prognosis. However, it is unexpected to switch from median CD47 H-scores as cut-off values for Kaplan-Meier analyses to Δ CD47 H-scores (post- vs. pre-chemotherapy paired specimens) for survival and death groups (Figure 1). How to discriminate between patients with stable high or stable low levels that would have similar Δ CD47 H-score but probably different outcome?

Convincing results suggest that the modulation CD47 levels in tumor cells is driven by other cells of the microenvironment. The orientation towards the investigation of soluble factors is not clear since a well-known ligand for CD47 is the protein signal regulatory protein α (SIRP α) on macrophages. What are the arguments to discard the possibility of a regulation through cell-cell contact? Does doxorubicin affect SIRP α levels on macrophages?

The Authors indicate: "IL18 was primarily expressed in the macrophages" (Line 219). Is it dependent on macrophage polarization (M0, M1, M2)? Does doxorubicin treatment of M1- or M2-pre-polarized macrophages stimulate IL18 secretion in a similar manner?

What is the LAT2 expression level in HOS cells treated with doxorubicin? Is the modulation restricted to IL18-dependent cascade?

Complementary correlation analyses could investigate in cohorts, as for example the relation between LAT2 and prognosis/survival; LAT2/IL18/CD47; or CD47/IL18/M1 macrophage markers.

---Minor comments

The clinical characteristics of the 81 patients included in this study have to be mentioned and discussed (age, gender, individual chemotherapy regimen and response, metastatic status...).

H-scores could be calculated for the different markers illustrated in Fig 2C and Fig S2G.

Figure S1A illustrates the absence of modulation in CD47 levels in tumor cells incubated for 24h with increasing dose of doxorubicin. In contrast, in the preclinical model, tumor-bearing mice received doxorubicin or solvent intravenously four times every 2 days. An in vitro time course study should be performed to fully confirm that doxorubicin do not directly affect CD47 levels in HOS cells. Similar dose- and time-dependent effect of doxorubicin on CD47 levels could be investigated in the other named osteosarcoma cell lines (U-2 OS, MG63, 143B).

The legend of Fig 1D should be checked as it indicates trypsinization of tumor samples before CD47 staining for flow cytometry analysis, whereas the mat&met section indicates collagenase IV/hyaluronidase dissociation.

The sequence of primers for beta-Actin are missing in Table S3. In contrast, some primers mentioned in the Table S3 are not used in the manuscript (itgam, mrc1).

The reference for Actin and GAPDH antibodies are missing.

Reviewer #4 (Remarks to the Author): with expertise in immunometabolism

This study revealed an intriguing mechanism underlying chemotherapy-induced macrophage phagocytosis suppression, which is mediated by IL-18-induced and LAT2-dependent CD47 upregulation in osteosarcoma, promoting immune evasion. However, several points remain unclear and are required investigation for further consideration.

Major:

1. The data demonstrated the polarization of tumor-associated macrophages (TAMs) from in vivo and in vitro were not clear to me. Especially the expression of CD206 in TAMs (Fig. 2A). Also, the markers for pro-inflammatory (M1) and anti-inflammatory (M2) macrophages are very limited and not

convincing at all. Thus, in addition to CD86 and CD206, markers such as Arg1, MHCII, and Nos2 would be great to put into this manuscript for audiences to have a clear idea of the macrophage activation state.

2. The signal of IL-18 in Fig. 3A is weak, which makes me wonder whether or not IL-18 is (directly) secreted by tumor macrophages during the doxorubicin treatment. Therefore, the authors have to perform the expt illustrated in Fig 2D using IL-18 deficient (IL-18 shRNA) macrophages to assess whether the expression of CD47, mTORC1 and c-myc is altered in the tumor cells.

3. The authors claimed that TAM secreted IL-18 is only induced by the doxorubicin treatment. If this is true, why the group of tumor-bearing mice treated with "saline+mIL-18BP" exhibited a marked reduction of tumor progression.

4. Does doxorubicin treatment induce the expression of SIRPa in tumor-associated macrophages?

5. In Fig. 3E, it is unclear why mIL-18BP can induce the population of CD86+ (M1) macrophage population compared to PBS? Similarly, why does shLAT2 or BCH cause macrophage infiltration (Fig. 6A and 6G)?

6. It is known that glutamine metabolism is essential for immunosuppressive (M2) macrophages. Do TAMs expression LAT2 in osteosarcoma? Does BCH treatment affect macrophage immunity and metabolism?

7. The role of c-myc to CD47 and immune evasion in osteosarcoma is not clear to me; thus, in addition to LAT2 shRNA, the authors need to perform c-myc deficient (shRNA) osteosarcoma with or without doxorubicin treatment in vivo expt.

8. Similarly, do c-myc overexpressed tumor cells overcome the Glu/Leu deprivation and LAT2 blockade (BCH treatment) to upregulate CD47 affecting macrophage phagocytosis?

Minor:

1. For all the histogram data from the flow cytometry analysis, isotype control is required.

2. In addition to IL-18, are other macrophages-produced cytokines as IL-1b, IL-6, TNF-a, IL-10 affected by the doxorubicin treatment?

3. In Fig. S1B, does the treatment of doxorubicin affect the % and IFNg production of CD4 and CD8 T cells?

4. In Fig 6c,d and h, the clodronate (alone) treatment control is missing. I am wondering that does clodronate (alone) treatment affect tumor progression?

Reviewer #1

In this study by Wang et al, the authors showed enhanced CD47 expression in osteosarcoma tissues after chemotherapy treatment. Further investigation by the authors demonstrated a role of IL18, secreted by macrophages in response to doxorubicin, in inducing LAT2 expression for leucine and glutamine uptake which in turn activated mTORC1 and c-Myc for enhancing CD47 expression. Their findings revealed important immune evasion mechanisms developed by osteosarcoma during chemotherapy and identified the underlying mechanisms for CD47 upregulation. This study was well performed with proper controls.

Despite the enthusiasm, there is a lack of convincing evidence proving some of the key conclusions and/or connecting the findings from different experiments, which need to be adequately addressed by the authors (detailed in below).

Answer: We greatly appreciate the reviewer's acknowledgement of the potential significance of our findings and the insightful comments, which are essential for improvement of this manuscript.

Major concerns:

1. In Fig3, the role of IL18 in doxorubicin-induced CD47 upregulation was examined by IL18BP. However, whether and how IL18BP functions as an antagonist of IL18 was not established. If IL18BP functions through competing with endogenous IL18 for binding to IL18 receptors on osteosarcoma cells, a binding assay should be performed to demonstrate such effects. In addition, additional IL18 blocking approaches should be used to confirm its role – eg. knockdown of IL18 expression in macrophages. Moreover, in vivo experiments with osteosarcoma cells deficient in responding to IL18 (eg. IL18R knockdown osteosarcoma cells) should be performed to demonstrate doxorubicin cannot induce CD47 upregulation in such cells.

Answer: It was reported that IL-18BP is a secreted decoy receptor, which binds IL-18 with an affinity significantly higher than that of IL-18R1, thereby inhibiting the interaction between

IL18 and IL-18R (PMID: 10023777, PMID: 10655506, PMID: 22427351). We have included this published information in our revised manuscript (**Page 10, Line 228**).

To further determine the role of IL18 in the current study, we performed the reviewer-suggested experiments with depletion of IL18 and IL18R1 in macrophages and tumor cells, respectively. We showed that depletion of IL18 in differentiated THP-1 cells suppressed doxorubicin-induced CD47 upregulation in HOS tumor cells (**Fig. 3H**) and enhanced macrophage-mediated phagocytosis of HOS and SJSA-1 cells (**Figs. 3I, Supplementary Fig. 3L**). Consistently, depletion of IL18R1 in HOS and SJSA-1 tumor cells (**Supplementary Fig. 3N**) greatly decreased both the basal and doxorubicin-upregulated CD47 expression (**Supplementary Figs. 3O-3R**) with corresponding increase of macrophages, activation of macrophages with M1-polarity (**Supplementary Figs. 3O-3R**) and inhibition of tumor growth (**Supplementary Figs. 3S and 3T**).

2. In Fig5, the authors claimed the activation of mTORC1 led to subsequent c-Myc-mediated transcription of CD47. However, their data only suggested IL18 induced c-myc expression and S6K phosphorylation which were abolished by rapamycin. There were not sufficient data showing enhanced CD47 expression was regulated by mTORC1 and c-Myc. This can be addressed by examining the effects of a knockdown of S6K and/or c-Myc on LAT2-mediated CD47 expression.

Answer: We showed in our original submission that treatment of c-Myc inhibitor 10058-F4 suppressed both the basal and IL-18-induced CD47 expression in HOS (**original Fig 5F, appeared as revised Fig. 6G**), U-2 OS (**original Supplementary Fig 4E, appeared as revised Supplementary Fig. 6F**), and SJSA-1 (**Figs. 6G**) cells and abolished IL18-suppressed phagocytosis (**Fig. 6I, Supplementary Fig. 6H**).

We also performed the reviewer-suggested experiments. We showed that c-Myc knockdown in HOS cells abolished IL-18-induced CD47 upregulation (**Supplementary Fig. 6I**) and phagocytosis inhibition (**Supplementary Fig. 6J**).

3. The clinical significance of combining doxorubicin and LAT2 inhibitors is unclear. As shown in Fig2F, although doxorubicin induced resistance to phagocytosis by increasing CD47 expression, this can be overcome by CD47 antibody which strongly induced phagocytosis of both untreated and doxorubicin-treated cells. Therefore, one would predict the upregulation of CD47 by doxorubicin would not compromise the efficacy of CD47 antibody, and a combination of doxorubicin and CD47 antibody treatment may induce stronger anti-cancer effects in osteosarcoma than that by doxorubicin+BCH. These should be examined with the in vivo models.

Answer: We showed that BCH treatment significantly reduced both basal and doxorubicin-upregulated CD47 expression (about 90% reduction) in tumor cells (**Figs. 5E, 7G and 7H, Supplementary Fig. 7E**). Thus, this dramatic drop of CD47 should have comparable effect on tumor growth to those induced by the CD47 antibody treatment. As expected, animal studies showed that BCH and the CD47 antibody treatment exhibited similar levels of tumor growth inhibition (**Supplementary Fig. 7I**). In addition, a combination of doxorubicin and CD47 antibody treatment induced comparable anti-cancer effects in osteosarcoma to that induced by combine treatment of doxorubicin with BCH (**Supplementary Fig. 7I**).

4. In the phagocytosis assays shown in Fig4F, Fig5G, etc., it's not convincing the changes of phagocytosis were due to up- or down-regulation of CD47. A control group of osteosarcoma cells (CD47 knockdown or anti-CD47 Fab treatment to block CD47) should be included to demonstrate the effects of IL18 and/or BCH/rapamycin on phagocytosis are abolished when CD47 is blocked.

Answer: We performed the review-suggested experiments. We showed that CD47 antibody treatment abrogated the IL18-suppressed phagocytosis (**Fig. 3C, Supplementary Fig. 3E**).

Minor issues:

1. Fig1, the experimental details of the in vivo experiments should be described;

Answer: We have added the detailed experimental procedures in Figure 1 legend. in our revised manuscript.

2. Fig2C, Fig3E, Fig6A, and FigS2G, the expression levels of the markers examined should be quantified.

Answer: We have added the quantification results for all IHC analysis in our revised manuscript (**Figs. 2G, 3D, 3K, 7B and 7H, Supplementary Figs. 3P and 3R**).

3. Fig2 and Fig3, the conditioned medium generated from THP1 cell line should not be termed TAM conditioned medium.

Answer: THP-1 cells were differentiated into macrophages by PMA before generation of conditioned medium. We have changed the term into the macrophage-conditioned medium (M-CM) in our revised manuscript.

Reviewer #2

In this manuscript by Wang et al., the authors present evidence that LAT2-mediated amino acid uptake controls doxorubicin-induced macrophage evasion by osteosarcoma cells. Using the human HOS cell line as a primary model, the authors demonstrate that doxorubicin treatment enhances the expression of the immune checkpoint protein CD47 on tumor cells *in vivo* and propose a macrophage-dependent mechanism. The authors extend this observation to a small osteosarcoma cohort of paired pre- and post-chemotherapy specimens, and found that patients with higher CD47 expression post-chemotherapy had shorter overall and disease-free survival. Using tumor-associated macrophage (TAM) conditioned media, the authors identified that doxorubicin promotes the secretion of IL-18 by macrophages. Supplementing tumor cells with IL-18 was sufficient to induce CD47 expression. In contrast, antagonizing IL-18 abrogated the increase in CD47 expression and enhanced phagocytosis of tumor cells by macrophages.

The authors subsequently showed that IL-18 promotes the expression of the amino acid transporter LAT2 in osteosarcoma cells, as well as uptake of the amino acids Gln and Leu. Genetic depletion of LAT2, pharmacological inhibition of the LAT proteins, or depletion of Gln/Leu in tumor cells diminished IL-18-induced expression of CD47 and enhanced phagocytosis by macrophages. The authors further showed that supplementing amino acid-starved tumor cells with Gln or Leu increased mTORC1 activity and c-Myc expression, which were abrogated by LAT2 depletion or inhibition. Treatment of tumor cells with rapamycin or a c-Myc inhibitor phenocopied LAT2 inhibition and diminished IL-18-induced CD47 expression. Finally, using a series of HOS xenograft experiments, the authors demonstrated that LAT2 depletion or inhibition enhances the efficacy of doxorubicin *in vivo*, which they claim is associated with increased M1 macrophage infiltration into the tumor.

The manuscript is clearly written, and the main conclusions have important implications for tumor immunology and immunotherapy. Nonetheless, there are a few issues that need to be addressed to fully support the authors' conclusions before this manuscript can be considered for publication.

Answer: We greatly appreciate the reviewer's acknowledgement of the potential significance of this report and the insightful comments. We have performed all necessary experiments to address the reviewer's concerns.

Major comments:

1. In the Abstract and throughout the manuscript, the authors claim that chemotherapy enhances the expression levels of CD47; however, only doxorubicin was evaluated in this study. Is the secretion of IL-18 by TAMs and the subsequent upregulation of CD47 by osteosarcoma cells specific to doxorubicin, or can other chemotherapies that are used to treat osteosarcoma elicit a similar response?

Answer: We performed additional experiments and examined the changes of IL-18 production and CD47 expression *in vitro* and *in vivo* in response to cisplatin, which is also used for osteosarcoma treatment. In the cisplatin-treated macrophage conditioned media (M-CM^{Cis/Cis}), IL-18 was substantially upregulated (**Supplementary Fig. 3G**). Consistently, HOS cells cultured in M-CM^{Cis/Cis} exhibited a considerable increase of CD47 expression (**Supplementary Fig. 3I**). The similar results were also obtained in mouse experiment showing that cisplatin increased the levels of IL-18 and CD47 in tumors (**Figs. 1B and 1C, Supplementary Figs. 1A and 3H**).

We also examined the effect of methotrexate and ifosfamide on IL-18 production and CD47 expression and did not detect obvious changes (**Figs. 1B and 1C, Supplementary Figs. 1A and 3G-3I**). Given that the current first-line osteosarcoma therapy is a combination of doxorubicin, cisplatin, and other chemotherapeutic drugs, it would reasonably conclude that the chemotherapy containing doxorubicin or cisplatin induces CD47 expression and promotes tumor immune evasion. We have modified the description and discussion in the text accordingly.

2. Throughout the manuscript, summarized flow cytometry data are presented (e.g., Figure 4F, 5B, 5G, S1B, etc.), but the corresponding flow cytometry plots are not shown. These plots should be provided as supplementary figures.

Answer: We have included all the flow cytometry plots in our revised manuscript (**Figs. 2A-2E, 2K, 2L, 3C, 3G, 3I and 5G, Supplementary Figs. 2I-2M, 2Q, 3E, 3L, 5F, 6B, 6G, 6H,**

6J, 6L, 6N, 6Q, 7B, 7F and 7G).

3. In Figure 2C, 3E, 6A, 6G, etc., the authors draw conclusions from qualitative IHC staining. Quantification of the staining (e.g., average number of positive cells per tumor area, pixel intensity, etc.) would significantly aid in data interpretation. For example, the authors use clodronate liposomes in the study to deplete macrophages and claim that this treatment reduces F4/80 immunostaining in tumors (Figure 2C); however, this conclusion is not supported by the small ‘representative’ images shown by the authors. Quantification of F4/80 and CD86 staining is required to support the authors’ claims.

Answer: We have added the quantification results for all IHC analysis in our revised manuscript (**Figs. 2G, 3D, 3K, 7B and 7H, Supplementary Figs. 3P and 3R**).

4. As an extension to #4, the authors claim that doxorubicin + LAT2 depletion (Figure 6A) or LAT2 inhibition (Figure 6G) enhances the recruitment of macrophages into the tumor. This conclusion is not supported by the IHC data. The authors should quantify the F4/80 and CD86 staining. In addition, the authors could quantify TAMs (total, M1, and M2) by flow cytometry, as the authors did in Figures 1 and S1, at an early time-point when tumor volumes are similar between treatment groups.

Answer: As suggested by the reviewer, we added the quantification results for F4/80 and CD86 staining (**Figs. 7B and 7H**) and performed flow cytometry analysis to quantify TAMs (**Figs. 7D and 7I, Supplementary Figs. 7B and 7F**). Both analyses showed similar results.

In line with previous report (PMID: 31376208), doxorubicin, which did not obviously change the total population of TAM, activated macrophages with M1-polarity. LAT2 depletion (**Figs. 7A, 7B and 7D, Supplementary Fig. 7B**) or inhibition (**Figs. 7G-7I, Supplementary Figs. 7F and 7G**) substantially increased both total and M1-polarized macrophages in absence or presence of doxorubicin.

5. In Figure 5, the authors demonstrate that treatment of HOS cells with either rapamycin (Figure 5E) or 10058-F4 (Figure 5F) impairs IL-18-induced CD47 expression; however, only

rapamycin was further evaluated and shown to enhance phagocytosis of HOS cells by macrophages. Does inhibition of c-Myc with 10058-F4 also enhance phagocytosis?

Answer: We performed the reviewer-suggested experiment and showed that 10058-F4 enhanced the macrophage phagocytosis and abrogated IL-18-induced phagocytosis suppression (**Fig. 6I, Supplementary Fig. 6H**).

6. In the Discussion (lines 331-334), the authors cite a study that reported HIF-dependent enrichment of CD47 in response to chemotherapy in triple-negative breast cancer cells and state that regulation of CD47 can be tumor type-dependent. Did the authors evaluate HIF expression in their osteosarcoma cells following doxorubicin treatment or evaluate common markers of hypoxia (e.g., pimonidazole) *in vivo*? Is this mechanism also evident in osteosarcoma cells?

Answer: In osteosarcoma cells, HIF-1 α and HIF-2 α expression levels were not obviously increased upon doxorubicin treatment (**Supplementary Fig. 9A**), suggesting HIF-mediated signaling is not evident in inducing CD47 upregulation.

7. In Supplementary Figure S2H, the authors demonstrate that mRNA expression of IL18R1 is significantly higher in osteosarcoma cell lines compared to a normal osteoblast cell line. Does doxorubicin treatment further increase the expression of IL18R1 in osteosarcoma cells?

Answer: The IL-18R1 expression remained unchanged upon doxorubicin treatment in HOS and SJSA-1 cells *in vivo* and *in vitro* (**Figs. R1A and R1B**). In addition, doxorubicin-treated macrophage conditioned medium (M-CM^{DOX/DOX}) did not alter IL-18R1 expression in both osteosarcoma cell lines (**Fig. R1C**).

Fig. R1. A, Representative immunohistochemical images showing IL-18R1 expression in serial sections of HOS (left) or SJSA-1 (right) tumors treated with saline or doxorubicin on day 22 or day 29. Scale bar, 50 μ m. Statistical analysis of quantification is shown on the right ($n=5$ mice per group). ns, not significant. Unpaired two-tailed Student t test. Results are presented as mean \pm SD. **B**, Western blot analysis of HOS (left) or SJSA-1 (right) treated with incremental doses of doxorubicin for 24 h. **C**, Western blot analysis of HOS (left) or SJSA-1 (right) cells treated with different macrophage conditioned media (M-CM) as indicated for 24 h.

8. A major limitation of the study is that the authors rely primarily on a single cell line (HOS cells). While other osteosarcoma lines are periodically assayed in the study, the authors should ensure that their main conclusions are supported by data from more than one cell line. For example, does LAT2 depletion or inhibition sensitize U-2 OS cells to doxorubicin in vivo?

Answer: We have performed all the key experiments including the reviewer-suggested ones using additional osteosarcoma cell lines (SJSA-1 and U-2 OS cells) and obtained similar results, as described below:

Doxorubicin-treated macrophage conditioned medium (M-CM^{DOX/DOX}) induced CD47 expression and inhibited phagocytosis of SJSA-1 cells (Figs. 2I, 2J and 2L). IL-18 inhibited phagocytosis of SJSA-1 cells by upregulating CD47 expression (Supplementary Figs. 3C-3E). IL-18 knockdown in macrophages abrogated doxorubicin-induced phagocytosis suppression (Supplementary Fig. 3L). IL18R1 knockdown in SJSA-1 cells inhibited doxorubicin-induced CD47 upregulation, enhanced macrophage infiltration and M1 polarization, and improved anti-tumor effect (Supplementary Figs. 3N, 3Q, 3R and 3T). LAT2 inhibition by BCH sensitized SJSA-1 tumors to doxorubicin (Supplementary Fig. 7D)

accompanying with reduced CD47 expression (**Supplementary Fig. 7E**) and enhanced macrophage infiltration and M1 polarization (**Supplementary Fig. 7G**). IL-18 upregulated CD47 expression in SJSA-1 cells (**Supplementary Figs. 3C and 3D**), and depletion of glutamine, leucine or both abrogated the IL-18-mediated CD47 upregulation (**Supplementary Fig. 6C**). BCH (**Supplementary Figs. 5D and 5F**), rapamycin (**Figs. 6F and 6H, Supplementary Fig. 6G**), and c-Myc inhibitor, 10058-F4 (**Figs. 6G and 6I, Supplementary Fig. 6H**) enhanced phagocytosis of SJSA-1 cells by reducing CD47 expression.

Minor comments:

1. Many figure legends do not state whether the error bars represent the standard error of the mean (SEM) or standard deviation (SD). This should be made clear for each figure, where appropriate.

Answer: The error bars for each figure were described clearly in our revised manuscript.

2. For the overlaid histograms in Figures 1D, 2A, and 2E, the data should be normalized to mode. Moreover, in Figure 2A, the quantification of MFI for CD206 is not appropriate, as two distinct populations are apparent. Reporting the percentage of CD206⁺ cells would be more appropriate.

Answer: All histograms were normalized to mode (**Figs. 1C and 2I, Supplementary Figs. 2S, 2T, 6O and 7E**). Flow cytometry results in original Figure 2A are presented with dot plot, and percentage of CD206⁺ macrophages are shown in our revised manuscript (**Fig. 2E**).

3. What gating strategy was used for Figure 2A? What is the parent population of the CD86⁺ and CD206⁺ cells? This should be reported in the figure and/or figure legend.

Answer: The gating strategy was shown in **Supplementary Fig. 8D** in our revised manuscript. The parent population of CD86⁺ and CD206⁺ cells was CD45⁺ CD11b⁺ F4/80⁺ macrophages, which were described in the figure legends of **Figs. 2A-2E** in our revised manuscript.

4. What markers were used to distinguish each cell type in Supplementary Figure S1B? This should be reported in the figure and/or figure legend.

Answer: Markers used to distinguish each cell type were described in the figure legends of **Supplementary Figs. 2I-2M.**

5. The authors use the word “synergistic” to describe the interaction between doxorubicin and BCH (line 307 and 373); however, the authors do not formally evaluate whether the interaction between doxorubicin and BCH is additive or synergistic. In the absence of this analysis, the authors should refrain from using the word “synergistic”.

Answer: We removed “synergistic” from the description and stated that BCH “sensitized” the tumor to doxorubicin treatment.

Reviewer #3

The present manuscript describes the induction of immunosuppression under chemotherapy in the context of osteosarcoma. The Authors highlighted the key role of the tumor microenvironment, notably through induction of IL18 secretion by the macrophages. The activation of IL18 receptor at the surface of tumor cells led to an upregulation of LAT2, the subsequent modulation of the amino acid metabolism (mainly Leu and Gln), and the activation of mTOR signaling pathway leading to the ignition of c-myc targeted promoter of the CD47 gene. The upregulation of CD47 at the tumor cell surface triggers the inhibition of macrophage phagocytosis, and thus correlates with tumor progression and worse prognosis.

The Authors used various models: modified HOS cell lines *in vitro* and *in vivo* (CDX models), osteosarcoma tissue samples from patients, and bioinformatics database (GEO).

Complementary approaches were used to fully confirm their hypotheses. The manuscript is well written, and the conclusions are in line with the results. However, in order for this paper to be fully suitable for publication, several minor points have to be considered.

Answer: We greatly appreciate the reviewer's acknowledgement of the potential significance of this report and the insightful comments. We have performed all necessary experiments to address the reviewer's concerns.

---Major comments

1. The Authors indicate: "patients received chemotherapy regimens consisting of doxorubicin, cisplatin, and high-dose methotrexate, and some of them additionally received ifosfamide" (Line 398), but tested only doxorubicin for the IL18/CD47 induction. What about other conventional drugs onto macrophage activation and crosstalk with tumor cells?

Answer: We performed additional experiments and examined the changes of IL-18 production and CD47 expression *in vitro* and *in vivo* in response to cisplatin, which is also used for osteosarcoma treatment. In the cisplatin-treated macrophage conditioned media (M-CM^{Cis/Cis}), IL-18 was

substantially upregulated (**Supplementary Fig. 3G**). Consistently, HOS cells cultured in M-CM^{Cis/Cis} exhibited a considerable increase of CD47 expression (**Supplementary Fig. 3I**). The similar results were also obtained in mouse experiment showing that cisplatin increased the levels of IL-18 and CD47 in tumors (**Figs. 1B and 1C, Supplementary Figs. 1A and 3H**).

We also examined the effect of methotrexate and ifosfamide on IL-18 production and CD47 expression and did not detect obvious changes (**Figs. 1B and 1C, Supplementary Figs. 1A and 3G-3I**). Given that the current first-line osteosarcoma therapy is a combination of doxorubicin, cisplatin, and other chemotherapeutic drugs, it would reasonably conclude that the chemotherapy containing doxorubicin or cisplatin induces CD47 expression and promotes tumor immune evasion. We have modified the description and discussion in the text accordingly.

2. Convincing results suggest that high CD47 levels correlate with poor prognosis. However, it is unexpected to switch from median CD47 H-scores as cut-off values for Kaplan-Meier analyses to Δ CD47 H-scores (post- vs. pre-chemotherapy paired specimens) for survival and death groups (Figure 1). How to discriminate between patients with stable high or stable low levels that would have similar Δ CD47 H-score but probably different outcome?

Answer: The point is well taken. Consistent with previous studies, patients with high CD47 H-scores pre-chemotherapy showed poorer prognosis than those with low CD47 H-scores (**Fig. 1D, Supplementary Fig. 1B**). We additionally found patients with high Δ CD47 H-scores showed poorer prognosis than those with low Δ CD47 H-scores (**Fig. 1E, Supplementary Fig. 1C**). We then stratified the patients into four groups based on CD47 H-scores and Δ CD47 H-scores (**Fig. 1F, Supplementary Fig. 1D**): CD47^{low} Δ CD47^{low}, CD47^{low} Δ CD47^{high}, CD47^{high} Δ CD47^{low}, and CD47^{high} Δ CD47^{high}. These optimal cutoff values for CD47 H-score and Δ CD47 H-score were determined by X-tile as described in methods and figure legends. We showed that the patients with CD47^{low} Δ CD47^{high} had poorer prognosis than the patients with CD47^{low} Δ CD47^{low} (**Supplementary Fig. 1D**). Similarly, the patients with CD47^{high} Δ CD47^{high} had poorer prognosis than the patients with CD47^{high} Δ CD47^{low} (**Fig. 1F, Supplementary Fig. 1D**). These results suggested that the levels of CD47 upregulation elicited by chemotherapy is associated with poorer prognosis of the patients.

3. Convincing results suggest that the modulation CD47 levels in tumor cells is driven by other cells of the microenvironment. The orientation towards the investigation of soluble factors is not clear since a well-known ligand for CD47 is the protein signal regulatory protein α (SIRP α) on macrophages. What are the arguments to discard the possibility of a regulation through cell-cell contact? Does doxorubicin affect SIRP α levels on macrophages?

Answer: To examine whether the contact between the tumor cells and macrophages regulates CD47, we cocultured tumor cells with macrophages either by seeding them into the same dish to allow their cell-cell contacts (direct coculture) or seeding macrophages into the upper insert of a transwell apparatus with tumor cells in the lower six-well plates (indirect coculture). We showed that doxorubicin treatment elicited similar levels of CD47 upregulation in these two conditions (**Supplementary Fig. 2S**), suggesting that doxorubicin-upregulated CD47 expression primarily results from secreted soluble factors by macrophages.

Doxorubicin treatment did not alter the SIRP α expression in THP-1-derived macrophages and BMDMs in the presence of DMSO- or doxorubicin-treated HOS conditioned medium (T-CM^{DMSO} or T-CM^{DOX}) (**Supplementary Fig. 9B**). However, doxorubicin treatment of mice increased SIRP α expression in tumor-associated macrophages (**Supplementary Fig. 9C**), suggesting that secreted soluble factors from macrophages-surrounding cells in tumor microenvironment enhances SIRP α expression. These results also suggested that both enhanced expression of CD47 in tumor cells and upregulated SIRP α in tumor-associated macrophages plays a role in tumor immune evasion and that disrupting CD47 upregulation is effective for sensitizing doxorubicin treatment.

4. The Authors indicate: “IL18 was primarily expressed in the macrophages” (Line 219). Is it dependent on macrophage polarization (M0, M1, M2)? Does doxorubicin treatment of M1- or M2-pre-polarized macrophages stimulate IL18 secretion in a similar manner?

Answer: We showed that compared to M0 macrophages, M1 and M2 macrophages had increased and decreased IL-18 secretion, respectively, in the absence of doxorubicin treatment (**Fig. R2A**).

Doxorubicin treatment (M2-CM^{DMSO/DOX}) did not alter IL-18 secretion of M2 macrophages. However, doxorubicin treatment combined with doxorubicin-treated tumor-conditioned medium (M2-CM^{DOX/DOX}) promoted IL-18 secretion from M2 macrophages (Fig. R3B, right). In contrast, both doxorubicin (M1-CM^{DMSO/DOX}) and doxorubicin-treated tumor-conditioned medium (M1-CM^{DOX/DMSO}) substantially increased IL-18 secretion of M1 macrophages, and this increase was further enhanced with tumor-conditioned medium combining doxorubicin treatment (M1-CM^{DOX/DOX}) (Fig. R3B, left). These results suggested that tumor cell-secreted soluble factors promote doxorubicin-induced IL-18 secretion from both M1 and M2.

Fig. R2. A and B, THP-1 cells were treated with 320 nM PMA for 6 h and then with PMA, PMA plus 20 ng/ml IFN- γ and 100 ng/ml LPS or PMA plus 20 ng/ml IL-13 and 20 ng/ml IL-4 for 18 h to obtain the M0, M1 and M2 phenotype, respectively. **A**, Elisa analysis of IL-18 concentrations in supernatants from M0, M1, and M2 macrophages. *** $p < 0.001$ and one-way ANOVA. Error bars represent SD of four independent experiments. **B**, Elisa analysis of IL-18 concentrations in M1 (left) and M2 (right) conditioned media produced with the same procedure as described in **Figure 2H**. HOS cells were treated with DMSO or doxorubicin to produce DMSO-treated tumor conditioned medium (T-CM^{DMSO}) or doxorubicin-treated tumor conditioned medium (T-CM^{DOX}). M1/ M2 macrophages were treated with the T-CM^{DMSO} or T-CM^{DOX} in the presence of DMSO or doxorubicin to produce four kinds of M1-CM/M2-CM: M1-CM^{DMSO/DMSO}/M2-CM^{DMSO/DMSO} was produced after treated with T-CM^{DMSO} and DMSO; M1-CM^{DMSO/DOX}/M2-CM^{DMSO/DOX} was produced after treated with T-CM^{DMSO} and doxorubicin; M1-CM^{DOX/DMSO}/M2-CM^{DOX/DMSO} was produced after treated with T-CM^{DOX} and DMSO; M1-CM^{DOX/DOX}/M2-CM^{DOX/DOX} was produced after treated with T-CM^{DOX} and doxorubicin. ns, not significant. * $p < 0.05$, *** $p < 0.001$ and one-way ANOVA. Error bars represent SD of four independent experiments.

5. What is the LAT2 expression level in HOS cells treated with doxorubicin? Is the modulation restricted to IL18-dependent cascade?

Answer: We showed that doxorubicin-treated macrophage conditioned medium (M-CM^{DOX/DOX}) induced LAT2 expression in HOS cells, which was completely reversed by the depletion of IL-18 depletion (**Supplementary Fig. 5G**), suggesting that doxorubicin-induced an IL-18-dependent LAT2 upregulation in HOS cells.

6. Complementary correlation analyses could investigate in cohorts, as for example the relation between LAT2 and prognosis/survival; LAT2/IL18/CD47; or CD47/IL18/M1 macrophage markers.

Answer: We showed in our original submission that there is a positive correlation between CD 47 and IL-18 (**original Figs. 3I and 3J, Supplementary Fig. 2L, appeared as Figs.4E, 4G and 4H in the revised manuscript**). We conducted additional correlation analyses as the reviewer suggested, and found that patients with high upregulation of LAT2 expression (Δ LAT2^{high}) showed worse overall survival and disease-free survival (**Fig. 5I**); the expression of LAT2 was positively correlated with expression of IL-18 (**Supplementary Fig. 5H**) and the expression of CD47 was positively correlated with expression of LAT2 (**Supplementary Fig. 5I**) in osteosarcoma specimens before and after chemotherapy; the changes of LAT2 expression increased by chemotherapy were also positively correlated with the changes of IL-18 (**Supplementary Fig. 5J**), and the changes of CD47 expression increased by chemotherapy were also positively correlated with the changes of LAT2 (**Supplementary Fig. 5K**); *CD47* expression positively correlates with M1 marker *CD80* (**Supplementary Fig. 2N**). Similarly, *CD47* and M1 abundance showed positive correlation (**Supplementary Fig. 2O**) and patients with higher *CD47* expression showed higher M1 abundance (**Supplementary Fig. 2P**).

---Minor comments

1. The clinical characteristics of the 81 patients included in this study have to be mentioned and discussed (age, gender, individual chemotherapy regimen and response, metastatic

status...).

H-scores could be calculated for the different markers illustrated in Fig 2C and Fig S2G.

Answer: The clinical characteristics of all the patients including the indicated 81 osteosarcoma patients are summarized in **Supplementary Table S1** (The chemotherapy responses of patients before 2020 were not available). H-scores have been calculated for the different markers illustrated in original **Fig. 2C and Supplementary Fig. 2G (appeared as Fig. 2G and Fig. 3D in the revised manuscript)**.

2. Figure S1A illustrates the absence of modulation in CD47 levels in tumor cells incubated for 24h with increasing dose of doxorubicin. In contrast, in the preclinical model, tumor-bearing mice received doxorubicin or solvent intravenously four times every 2 days. An in vitro time course study should be performed to fully confirm that doxorubicin do not directly affect CD47 levels in HOS cells. Similar dose- and time-dependent effect of doxorubicin on CD47 levels could be investigated in the other named osteosarcoma cell lines (U-2 OS, MG63, 143B).

Answer: CD47 expression in different osteosarcoma cell lines (U-2 OS, MG63, HOS, and SJSA-1) remained unchanged after treating with doxorubicin for a wide range of time and doses (**Supplementary Figs. 2A-2H**).

3. The legend of Fig 1D should be checked as it indicates trypsinization of tumor samples before CD47 staining for flow cytometry analysis, whereas the mat&met section indicates collagenase IV/hyaluronidase dissociation.

Answer: The mistake has been corrected. Collagenase IV/hyaluronidase were used in the experiments.

4. The sequence of primers for beta-Actin are missing in Table S3. In contrast, some primers mentioned in the Table S3 are not used in the manuscript (itgam, mrc1).

Answer: We have included the sequence of primers for beta-Actin in the revised

Supplementary Table 4. *Itgam* and *Mrc1* are aliases for *Cd11b* and *Cd206*, respectively, which are used and shown in **Fig. 2F**. We have included this information in the revised **Supplementary Table 4**.

5. The reference for Actin and GAPDH antibodies are missing.

Answer: The references for actin and GAPDH antibodies were included in our revised manuscript (**Page 25, Line 605**).

Reviewer #4

This study revealed an intriguing mechanism underlying chemotherapy-induced macrophage phagocytosis suppression, which is mediated by IL-18-induced and LAT2-dependent CD47 upregulation in osteosarcoma, promoting immune evasion. However, several points remain unclear and are required investigation for further consideration.

Answer: We greatly appreciate the reviewer's acknowledgement of the potential significance of this report and the insightful comments. We have performed all necessary experiments to address the reviewer's concerns.

Major:

1. The data demonstrated the polarization of tumor-associated macrophages (TAMs) from in vivo and in vitro were not clear to me. Especially the expression of CD206 in TAMs (Fig. 2A). Also, the markers for pro-inflammatory (M1) and anti-inflammatory (M2) macrophages are very limited and not convincing at all. Thus, in addition to CD86 and CD206, markers such as Arg1, MHCII, and Nos2 would be great to put into this manuscript for audiences to have a clear idea of the macrophage activation state.

Answer: We thank the reviewer for raising this great point. As suggested, additional markers, such as Arg1, MHC-II, and NOS2, were examined by flow cytometry (Figs. 2A-2F, 7D and 7I, Supplementary Figs. 6Q, 7B, 7F and 7G), IHC analysis (Figs. 3J and 3K, Supplementary Figs. 3O-3R) and PCR analysis (Fig. 2F) in the revised manuscript. Consistent results were obtained.

2. The signal of IL-18 in Fig. 3A is weak, which makes me wonder whether or not IL-18 is (directly) secreted by tumor macrophages during the doxorubicin treatment. Therefore, the authors have to perform the expt illustrated in Fig 2D using IL-18 deficient (IL-18 shRNA) macrophages to assess whether the expression of CD47, mTORC1 and c-myc is altered in the tumor cells.

Answer: We performed a single-cell sequencing of osteosarcoma from patients and showed that IL18 was mostly expressed in macrophages (**Fig. 4B**). This result is also consistent with a previous report showing that IL-18 precursor is constitutively expressed in macrophages (PMID: 24115947).

As suggested by the reviewer, we depleted IL-18 in macrophage and showed that IL-18 knockdown abrogated mTORC1 activation, c-MYC and CD47 upregulation in HOS cells induced by doxorubicin-treated macrophage conditioned medium (**Fig. 3H, Supplementary Fig. 6D**). In addition, IL-18 depletion, abrogated the phagocytosis inhibition induced by doxorubicin-treated macrophage conditioned media in HOS (**Fig. 3I**) and SJSA-1 cells (**Supplementary Fig. 3L**). Furthermore, IL18R1 depletion in combination of doxorubicin treatment significantly increased infiltration of macrophages and activated macrophages with M1-polarity compared to doxorubicin treatment alone (**Supplementary Figs. 3O-3R**) and resulted in significantly inhibition of tumor growth (**Supplementary Figs. 3S and 3T**).

3. The authors claimed that TAM secreted IL-18 is only induced by the doxorubicin treatment. If this is true, why the group of tumor-bearing mice treated with “saline+mrIL-18BP” exhibited a marked reduction of tumor progression.

Answer: We apologized for unclear description and have revised it in the manuscript. What we showed is IL-18 has a basal expression from macrophages, which was enhanced about 2.5 folds by doxorubicin treatment (**Figs. 3A, 3D and 3E**). In addition, IL18 expression was positively correlated with CD47 expression in four cohorts of pre-chemotherapy osteosarcoma specimens (**Figs. 4E and 4G**). Consistent results showed that IL18R1 depletion (**Supplementary Figs. 3N-3T**), which mimicked IL-18BP treatment (**Figs. 3J-3L**), significantly reduced basal CD47 expression level and tumor growth without doxorubicin treatment. Thus, both basal and doxorubicin-enhanced IL18 secretion contribute to the promotion of CD47 upregulation and tumor growth.

4. Does doxorubicin treatment induce the expression of SIRP α in tumor-associated macrophages?

Answer: Doxorubicin treatment did not alter the SIRP α expression in THP-1-derived macrophages and BMDMs in the presence of DMSO- or doxorubicin-treated HOS conditioned medium (T-CM^{DMSO} or T-CM^{DOX}) (**Supplementary Fig. 9B**). However, doxorubicin treatment of mice increased SIRP α expression in tumor-associated macrophages (**Supplementary Fig. 9C**), suggesting that secreted soluble factors from macrophages-surrounding cells in tumor microenvironment enhances SIRP α expression. These results also suggested that both enhanced expression of CD47 in tumor cells and upregulated SIRP α in tumor-associated macrophages plays a role in tumor immune evasion and that disrupting CD47 upregulation is effective for sensitizing doxorubicin treatment.

5. In Fig. 3E, it is unclear why mrIL-18BP can induce the population of CD86+ (M1) macrophage population compared to PBS? Similarly, why does shLAT2 or BCH cause macrophage infiltration (Fig. 6A and 6G)?

Answer: As described in the answer to Point 3, IL-18 has a basal expression from macrophages and neutralization of IL-18 (**Figs. 3J-3L**) and depletion of IL18R1 (**Supplementary Figs. 3N-3T**) reduced basal CD47 expression levels and tumor growth without doxorubicin treatment. Similarly, shLAT2 and BCH also reduced CD47 expression in tumor cells (**Figs. 5C-5E, 7A, 7B, 7G and 7H, Supplementary Figs. 5D, 5E and 7E**). Thus, reduction of basal CD47 expression by mrIL-18BP, shLAT2, and BCH activates macrophages towards M1 phenotype and promotes macrophage infiltration. Our results are consistent with the publications showing that CD47 antibody monotherapy for osteosarcoma enhanced macrophage infiltration and M1 polarization (PMID: 35027753, PMID: 30674867, PMID: 31376208).

6. It is known that glutamine metabolism is essential for immunosuppressive (M2) macrophages. Do TAMs expression LAT2 in osteosarcoma? Does BCH treatment affect macrophage immunity and metabolism?

Answer: We performed additional mouse experiments and showed that LAT2 expression in tumor-associated macrophages was much lower than that in tumor cells in osteosarcoma tissues (**Supplementary Fig. 9D**). The barely detectable LAT2 expression in TAMs suggested that the effect of the BCH on macrophages is minimal. Consistently, BCH treatment did not directly influence the macrophage phagocytosis of tumor cells in vitro (**Supplementary Fig. 9E**). These results suggested that BCH treatment does not obviously alter the function of macrophages directly.

To examine whether BCH treatment affects macrophage metabolism, we performed metabolomics analysis and did not observe obvious alteration of glutamine metabolism of bone marrow-derived macrophages (BMDMs) with or without BCH treatment (**Table R1**). In addition, the abundance of glutamine in tumor associated macrophages (TAMs) from the tumor tissues is relatively low and below the detection limit. These results are consistent with a previous publication, which shows that TAMs primarily use glycolysis for metabolic needs (Nature. 2021; 593(7858): 282–288; PMID: 33828302).

Table R1. Significant KEGG pathways related to glutamine metabolism. Metabolomics data of BMDMs treated with or without BCH for 24 h were used for KEGG enrichment analysis.

Pathway	NumberCompound	NumberFeature	Background	Pvalue	FDR
D-Glutamine and D-glutamate metabolism	1	2	13	0.1217	0.1510

7. The role of c-myc to CD47 and immune evasion in osteosarcoma is not clear to me; thus, in addition to LAT2 shRNA, the authors need to perform c-myc deficient (shRNA) osteosarcoma with or without doxorubicin treatment in vivo expt.

Answer: We performed the reviewer-suggested experiments. c-Myc depletion greatly blocked doxorubicin-induced CD47 upregulation in tumor tissues (**Supplementary Fig. 6O**).

Consistently, compared with doxorubicin monotherapy or c-Myc knockdown alone, combination of c-Myc depletion with doxorubicin treatment greatly blunted tumor growth (**Supplementary Fig. 6P**) with increased infiltration of macrophages and polarization of M1 phenotype (**Supplementary Fig. 6Q**).

8. Similarly, do c-myc overexpressed tumor cells overcome the Glu/Leu deprivation and LAT2 blockade (BCH treatment) to upregulate CD47 affecting macrophage phagocytosis?

Answer: As expected, we found that c-Myc overexpression bypassed the Glu/Leu deprivation- and LAT2 blockade (BCH treatment)-dependent regulation, leading to upregulated CD47 expression (**Supplementary Figs. 6K and 6M**) and suppressed macrophage phagocytosis (**Supplementary Figs. 6L and 6N**).

Minor:

1. For all the histogram data from the flow cytometry analysis, isotype control is required.

Answer: We have included isotype control for all histogram data (**Figs. 1C and 2I, Supplementary Figs. 2S, 2T, 6O and 7E**).

2. In addition to IL-18, are other macrophages-produced cytokines as IL-1b, IL-6, TNF-a, IL-10 affected by the doxorubicin treatment?

Answer: We examined the effect of doxorubicin treatment on other macrophages-produced cytokines and found that IL-1 β , IL-6, and TNF- α production were not obviously enhanced by doxorubicin treatment (**Fig. R3**). IL-10 was not detected in our analyses (**Fig. R3**).

Fig. R3. Analysis of cytokines in M-CM^{DMSO/DMSO} and M-CM^{DOX/DOX} using a human cytokine array.

3. In Fig. S1B, does the treatment of doxorubicin affect the % and IFN γ production of CD4 and CD8 T cells?

Answer: Because nude balb/c mice lacking T cells were used in this study, we were unable to examine the effect of doxorubicin on the percentage and IFN-gamma production of CD4 and CD8 T cells. Deng C et al. demonstrated neoadjuvant chemotherapy was associated with increased densities of CD8⁺ T cells, Ki67⁺ CD8⁺ T cells analyzed by paired osteosarcoma pre- and post-chemotherapy specimens (PMID: 32232912).

4. In Fig 6c,d and h, the clodronate (alone) treatment control is missing. I am wondering that does clodronate (alone) treatment affect tumor progression?

Answer: We have added the clodronate (alone) treatment control, which did not affect tumor progression (Figs. 7E and 7J, Supplementary Fig. 7H).

REVIEWERS' COMMENTS

Reviewer #1 (Remarks to the Author):

The authors have adequately addressed all of my concerns. Thank you!

Reviewer #2 (Remarks to the Author):

The revised manuscript has addressed all concerns raised during the initial review. Some additional points:

- 1) On Line 159, the authors stated that doxorubicin reduced iNOS-positive macrophages in the tumor, but Figure 2B shows increased iNOS-positive macrophages.
- 2) The representative images in Figure 2G are difficult to see and interpret. Is it possible to replace these images with higher resolution and/or higher magnification insets. Arrows indicating positive cells would also aid in interpretation.

Reviewer #3 (Remarks to the Author):

The Authors fully addressed the comments. The revised manuscript is suitable for publication.

Reviewer #4 (Remarks to the Author):

Re Q5: Given that IL-18BP can induce CD86 (M1) macrophage, why DOX treatment activating IL-18-IL-18r also increases CD86 expression (Fig. 2C)? Please explain the mechanism underlying these two processes.

Reviewer #1 (Remarks to the Author):

The authors have adequately addressed all of my concerns. Thank you!

Reviewer #2 (Remarks to the Author):

The revised manuscript has addressed all concerns raised during the initial review. Some additional points:

1) On Line 159, the authors stated that doxorubicin reduced iNOS-positive macrophages in the tumor, but Figure 2B shows increased iNOS-positive macrophages.

Answer: We apologized for the mistake and changed it into “doxorubicin treatment upregulated M1 marker MHC-II, inducible nitric oxide synthase (iNOS) and CD86” in the revised manuscript (**Line 158**).

2) The representative images in Figure 2G are difficult to see and interpret. Is it possible to replace these images with higher resolution and/or higher magnification insets. Arrows indicating positive cells would also aid in interpretation.

Answer: We thank the reviewer for the useful suggestion. We have replaced with higher resolution images with arrows indicating positive cells (**Fig. 2g**).

Reviewer #3 (Remarks to the Author):

The Authors fully addressed the comments. The revised manuscript is suitable for publication.

Reviewer #4 (Remarks to the Author):

Re Q5: Given that IL-18BP can induce CD86 (M1) macrophage, why DOX treatment activating IL-18-IL-18r also increases CD86 expression (Fig. 2C)? Please explain the mechanism underlying these two processes.

Answer: IL-18BP treatment and DOX treatment induce CD86 (M1) macrophage with

distinct mechanisms: (1) Macrophage-mediated phagocytosis of tumor cells is able to activate macrophages with M1 phenotype (PMID: 30674867). IL-18BP treatment reduced both basal and DOX-induced CD47 expression on osteosarcoma cells to enhance macrophage-mediated phagocytosis of tumors cells, which resulted in increased macrophage CD86 expression (M1 polarization). (2) DOX treatment induced macrophage CD86 expression (M1 polarization) was mainly attributed to tumor-released damage-associated molecular patterns (DAMPs) (PMID: 33243969). DOX treatment induces tumor cell immunogenic cell death (ICD), causing the exposure and release of numerous DAMPs by tumor cells. DAMPs such as HMGB1 and ATP released by tumor cells can activate macrophages with pro-inflammatory phenotype, which is accompanied with increased CD86 expression. In addition, DAMP such as calreticulin exposed on tumor cell surface is a prophagocytic “eat me” signal, promoting macrophage phagocytosis (PMID: 33243969; PMID: 31376208).

Based on our observations in current study, DOX treatment not only induced prophagocytic “eat me” signal, but also enhanced CD47-mediated “don’t eat me” signal. DOX treatment in combined with IL-18BP treatment, which abrogated CD47-mediated “don’t eat me” signal, was able to further induce CD86 expression (M1 polarization), compared to DOX or IL-18BP monotherapy (Fig. 5a, b in the revised manuscript, Fig. R1b).

Fig. R1
a